

Atmospheric
Chemistry
and Physics

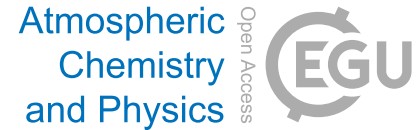

Research article

# Quantification of regional net $CO_2$ flux errors in the Orbiting Carbon Observatory-2 (OCO-2) v10 model intercomparison project (MIP) ensemble using airborne measurements

**Jeongmin Yun[1], Junjie Liu[1], Brendan Byrne[1], Brad Weir[2,3], Lesley E. Ott[2], Kathryn McKain[4], Bianca C. Baier[4], Luciana V. Gatti[5], and Sebastien C. Biraud[6]**

[1]Jet Propulsion Laboratory, California Institute of Technology, Pasadena, California, USA
[2]NASA Goddard Space Flight Center, Greenbelt, Maryland, USA
[3]Goddard Earth Sciences Technology and Research II, Morgan State University, Baltimore, Maryland, USA
[4]NOAA Global Monitoring Laboratory, Boulder, Colorado, USA
[5]General Coordination of Earth Science (CGCT), National Institute for Space Research (INPE),
São José dos Campos, Brazil
[6]Lawrence Berkeley National Laboratory, Berkeley, California, USA

**Correspondence:** Jeongmin Yun (jeongmin.yun@jpl.nasa.gov) and Junjie Liu (junjie.liu@jpl.nasa.gov)

**Abstract.** Inverse model intercomparison projects (MIPs) provide a chance to assess the uncertainties in inversion estimates arising from various sources. However, accurately quantifying ensemble $CO_2$ flux errors remains challenging and often relies on the ensemble spread. This study proposes a method for quantifying the errors in regional net surface–atmosphere $CO_2$ flux estimates from models taken from the Orbiting Carbon Observatory-2 (OCO-2) v10 MIP by using independent airborne $CO_2$ measurements for the period 2015–2017. We first calculate the root mean square error (RMSE) between the ensemble mean of posterior $CO_2$ concentrations and airborne observations and then isolate the $CO_2$ concentration errors caused solely by the ensemble mean of posterior net fluxes by subtracting the observation, representation, and transport errors from seven regions. Our analysis reveals that the flux errors projected onto $CO_2$ space account for 55 %–85 % of the regional average RMSE over the 3 years, ranging from 0.88 to 1.91 ppm. In five regions, the error estimates based on observations exceed those computed from the ensemble spread of posterior fluxes by a factor of 1.33–1.93, implying an underestimation of the actual flux errors, while their magnitudes are comparable in two regions. The adjoint sensitivity analysis identifies that the underestimation of flux errors is prominent where the magnitudes of fossil fuel emissions exceed those of terrestrial-biosphere fluxes by a factor of 3–31 over the 3 years. This suggests the presence of systematic biases in the inversion estimates associated with errors in the prescribed fossil fuel emissions common to all models. Our study emphasizes the value of airborne measurements for quantifying regional errors in ensemble net $CO_2$ flux estimates.

## 1 Introduction

Atmospheric CO$_2$ inverse modeling is a widely employed approach for estimating net surface–atmosphere CO$_2$ fluxes by assimilating observed atmospheric CO$_2$ concentrations. Most inverse-modeling approaches are based on Bayesian theory, wherein posterior flux is estimated from prior knowledge and atmospheric CO$_2$ observations, weighted by their uncertainties. This approach estimates a posterior probability distribution that can be represented as a maximum a posteriori solution (referred to as $\hat{x}$) and an error covariance matrix, following the notation of Rodgers (2000). Theoretically, since atmospheric CO$_2$ observations generally have a lower uncertainty than prior flux estimates, incorporating more observations causes the posterior fluxes to approach the true values (Liu et al., 2014).

However, concerns have been raised that inverse-modeling results are sensitive to the selection of transport models, prior flux datasets, and data assimilation techniques that are not accounted for in the Bayesian framework (Basu et al., 2018; Philip et al., 2019; Schuh et al., 2019). In order to obtain more robust flux estimates and assess their uncertainties, resulting from various sources (e.g., atmospheric transport and assimilation techniques), inverse model intercomparison projects (MIPs) have been conducted. These projects include the TransCom project (Gurney et al., 2004; Houweling et al., 2015), which was first initiated in the 1990s, as well as subsequent projects, such as the Global Carbon Project (GCP; Ciais et al., 2022; Friedlingstein et al., 2023) and the Orbiting Carbon Observatory-2 (OCO-2) MIP (Crowell et al., 2019; Peiro et al., 2022; Byrne et al., 2023). These MIPs involve different inverse-modeling groups using state-of-the-art transport modeling and assimilation techniques that assimilate in situ and satellite CO$_2$ data. Through these MIPs, researchers have analyzed differences in the maximum a posteriori solution across models. The OCO-2 MIP has revealed general agreement with regard to global flux estimates among ensemble models but significant discrepancies in regional fluxes, regardless of whether in situ and/or satellite data are assimilated (Crowell et al., 2019; Peiro et al., 2022).

Realistic error quantification of posterior fluxes from atmospheric flux inversions is essential for understanding how well regional fluxes are constrained by the current CO$_2$ observing network and for identifying regions with high uncertainty, allowing us to prioritize efforts to mitigate the error. The Bayesian formulation provides a method for calculating uncertainties in posterior fluxes based on uncertainties in prior fluxes and assimilated data. These uncertainties can be calculated analytically or approximated using a Monte Carlo method for variational methods (Chevallier et al., 2007; Feng et al., 2009; Liu et al., 2014); however, this is often computationally prohibitive for many inversion systems. This Bayesian posterior uncertainty accounts for random errors in the prior fluxes and observations but does not explicitly incorporate systematic errors, thus providing a potential underestimation of the total posterior error.

Errors in the maximum a posteriori fluxes are also commonly characterized through comparisons between independent atmospheric CO$_2$ measurements and posterior atmospheric CO$_2$ (Houweling et al., 2015; Crowell et al., 2019; Byrne et al., 2023). This approach can provide insights into the biases of current inverse modeling at global, latitudinal, or site-specific scales. However, as atmospheric CO$_2$ concentrations are influenced by both local and remote sources, it is difficult to identify regions where the observation–model comparison results are representative. Furthermore, these comparisons include not only posterior flux errors but also errors arising from transport, representation, and measurement. Because of these limitations, regional posterior flux errors in the ensemble mean have generally been defined as the ensemble spread among ensemble posterior fluxes. However, this method does not have an observational and theoretical basis and may not reflect actual errors (Byrne et al., 2023).

This study aims to develop a framework for quantifying errors in regional net surface–atmosphere CO$_2$ fluxes (terrestrial-biosphere fluxes + fossil fuel emissions) estimated from an ensemble of inverse models by using airborne CO$_2$ measurements, transport modeling, and an adjoint sensitivity analysis. Our target ensemble results are derived from 10 ensemble members of the OCO-2 v10 MIP for the period 2015–2017, which provide both posterior CO$_2$ fluxes and posterior CO$_2$ concentrations sampled at observation sites and times. The ensemble assimilates OCO-2 column-averaged dry-air mole fraction (XCO$_2$) retrievals (v10; O'Dell et al., 2018) and in situ CO$_2$ measurements (Tohjima et al., 2005; Nara et al., 2017; Schuldt et al., 2021a, b). This study uses more than 833 000 airborne CO$_2$ measurements collected at altitudes of 1–5 km above ground level (a.g.l.) from 20 different measurement projects (e.g., NOAA Carbon Cycle Group ObsPack Team, 2018; Baier et al., 2021; Miller et al., 2021; Schuldt et al., 2021a, b). These data have a broader spatial coverage and are less influenced by local sources compared to surface CO$_2$ data, thereby capturing signals from regional surface CO$_2$ fluxes. We quantify the errors in ensemble mean estimates of posterior atmospheric CO$_2$ by comparing them with the airborne CO$_2$ data. We then estimate the contributions of various error components (e.g., representation, observation, transport, and flux errors) to the observation–model difference in atmospheric CO$_2$ and isolate the contribution of flux errors. Next, we identify the areas to which these airborne CO$_2$ measurements are most sensitive and quantify the annual net flux errors in these areas.

## 2 Data and methodology

The aim of this study is to quantify the true errors in the ensemble net surface–atmosphere $CO_2$ fluxes generated by the OCO-2 v10 MIP using airborne observations. Here, the term "error" refers to the magnitude of the differences between the true and estimated flux values, without considering the sign. To achieve this, we employ three steps of analysis, as described in Fig. 1. First, we define two quantities: (1) the root mean square error (RMSE) between the ensemble mean of posterior $CO_2$ concentrations and observed $CO_2$ concentrations and (2) $ERR_{TOT}$ (Sect. 2.3). $RMSE^2$ represents the true error in the OCO-2 MIP ensemble mean of $CO_2$ concentrations, including the representation error ($\sigma_r^2$), observation error ($\sigma_o^2$), true flux error projected onto $CO_2$ concentrations ($\sigma_{f_t}^2$), transport error ($\sigma_t^2$), and error covariance between the preceding two terms ($\text{cov}(\sigma_{f_t}, \sigma_t)$). $ERR_{TOT}$ is the sum of the estimated error components, defined as the sum of $ERR_{REP}^2$, $ERR_{OBS}^2$, and $ERR_{MIP}^2$. $ERR_{REP}^2$ and $ERR_{OBS}^2$ denote the representation error ($\sigma_r^2$) and observation error ($\sigma_o^2$), respectively. $ERR_{MIP}^2$ represents the sum of the estimated flux error projected onto $CO_2$ space ($\sigma_{f_e}^2$) and the transport error ($\sigma_t^2$), along with the corresponding error covariance ($\text{cov}(\sigma_{f_e}, \sigma_t)$), computed from an ensemble spread of posterior $CO_2$ concentrations. Here, we separate representation errors from transport errors for computational purposes. The ratio between $ERR_{TOT}$ and the RMSE is then used to evaluate whether the estimated flux errors, computed from the ensemble spread of posterior fluxes, overestimate or underestimate the true errors in the ensemble mean fluxes. Next, we calculate the estimated flux error projected onto atmospheric $CO_2$ ($h(\text{err}_{f_e})$) through atmospheric transport simulations (Sect. 2.4). With $h(\text{err}_{f_e})$, $ERR_{TOT}$, and the RMSE, we derive the true error in the ensemble mean of posterior fluxes projected onto $CO_2$ space ($h(\text{err}_{f_t})$). Then, we identify the areas to which these airborne observations are most sensitive using an adjoint sensitivity analysis and calculate the estimated posterior flux error ($\text{err}_{f_e}$) over these regions. Assuming a linear observation operator, the study finally computes the true error in the ensemble mean posterior fluxes over the identified sensitive area ($\text{err}_{f_t}$) by applying the ratio between $h(\text{err}_{f_t})$ and $h(\text{err}_{f_e})$ to $\text{err}_{f_e}$.

### 2.1 The OCO-2 v10 MIP datasets

The OCO-2 v10 MIP provides multiple results from inverse models that assimilate different combinations of atmospheric $CO_2$ measurements for the period 2015–2020. Our study focuses on the results from the "LNLGIS" experiment, which assimilates the most observations except with respect to the OCO-2 ocean glint $XCO_2$ retrievals, which cause significant biases in inversion results (Byrne et al., 2023). The LNLGIS experiment incorporates OCO-2 v10 land nadir (LN) and land glint (LG) $XCO_2$ re-

trievals, along with global in situ (IS) data (including surface, ship-based, and airborne measurements) taken from obspack_co2_1_OCO2MIP_v3.2.1_2021-09-14. A total of 10 different inverse-modeling groups provided monthly posterior surface $CO_2$ flux estimates interpolated to a $1° \times 1°$ horizontal resolution and co-sampled posterior atmospheric $CO_2$ data corresponding to the times and locations of all types of observations. All of the inversion groups used the same fossil fuel emission estimates, based on the Open-source Data Inventory for Anthropogenic $CO_2$ (ODIAC) dataset (Basu and Nassar, 2021), but they independently chose their transport models, assimilation techniques, and prior flux estimates. Further details are provided in Table S1 in the Supplement, and more detailed explanations for each inverse-modeling approach can be found in Byrne et al. (2023). Although the OCO-2 MIP provides data for the period 2015–2020, we use data from the first 3 years due to the limited number of airborne measurements available during the later years. To minimize the influence of local sources and maximize the influence of regional fluxes, we exclude surface measurements and only consider airborne measurements taken between 1 and 5 km a.g.l. In addition, only airborne measurement data that were not assimilated in the LNLGIS experiment are used for analysis.

### 2.2 Airborne $CO_2$ measurement data

Figure 2a shows the spatial distribution of the total number of airborne $CO_2$ measurements used in this study within each $1° \times 1°$ grid cell. The dataset includes data from two airborne measurement campaigns conducted over the ocean – the Atmospheric Tomography Mission (ATom; Thompson et al., 2022) and the $O_2/N_2$ Ratio and $CO_2$ Airborne Southern Ocean (ORCAS) study (Stephens et al., 2018) – as well as 18 campaigns conducted over land. Specific airborne campaigns and their references are detailed in Table 1. The majority of the datasets used in the study are from North America, accounting for 37 % of the total observations for the period 2015–2017, followed by East Asia (35 %) and Alaska (7 %). The duration and extent of the airborne observations vary across different regions and time periods. Figure 2b illustrates the number of $1° \times 1°$ grid points in each of the seven regions where more than 10 observations are available per month. For Alaska, observations were concentrated during the Arctic–Boreal Vulnerability Experiment (ABoVE) campaign in 2017 (Sweeney et al., 2022). North America had observations for most of the analysis period, including observations from the Atmospheric Carbon and Transport – America (ACT-America) campaign, covering the eastern United States (Davis et al., 2021). The long-term Comprehensive Observation Network for TRace gases by AIrLiner (CON-TRAIL; Machida et al., 2008) project provides sparse observations for Europe and continuous observations for East and Southeast Asia for the period 2015 to 2017, as well as for Australia for 2015–2016. In South America, measurements

## 1) Evaluation of posterior flux error estimates over the globe

$$\underbrace{\frac{(\overbrace{ERR^2_{OBS}}^{\sigma^2_o} + \overbrace{ERR^2_{REP}}^{\sigma^2_r} + \overbrace{ERR^2_{MIP}}^{\sigma^2_{f_e} + cov(\sigma_{f_e},\sigma_t) + \sigma^2_t})_{(ppm)}}{RMSE^2_{\,(ppm)}}}_{\sigma^2_o + \sigma^2_r + \sigma^2_{f_t} + cov(\sigma_{f_t},\sigma_t) + \sigma^2_t} = \frac{ERR^2_{TOT\,(ppm)}}{RMSE^2_{\,(ppm)}} = \mathbf{Ratio^2} \begin{cases} > 1 : overestimated\ flux\ error \\ < 1 : underestimated\ flux\ error \end{cases}$$

## 2) Quantification of true flux errors by region

$$\overbrace{h\big(err_{f_t}\big)^2}^{\sigma^2_{f_t}} - \overbrace{h\big(err_{f_e}\big)^2}^{\sigma^2_{f_e}} \approx \underbrace{\overbrace{RMSE^2 - ERR^2_{TOT}}^{\sigma^2_o + \sigma^2_r + \sigma^2_{f_t} + cov(\sigma_{f_t},\sigma_t) + \sigma^2_t}}_{\sigma^2_o + \sigma^2_r + \sigma^2_{f_e} + cov(\sigma_{f_e},\sigma_t) + \sigma^2_t}$$

$$\frac{h\big(err_{f_e}\big)_{(ppm)}}{h\big(err_{f_t}\big)_{(ppm)}} = \frac{err_{f_e\ (gC\ m^{-2}\ day^{-1})}}{err_{f_t\ (gC\ m^{-2}\ day^{-1})}}$$

### $h\big(err_{f_e}\big)$ =

1. Simulate atmospheric CO$_2$ fields using forward modeling by prescribing posterior CO$_2$ fluxes for each ensemble member.

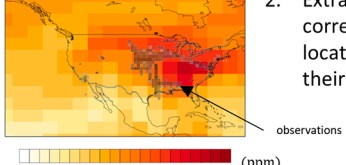

2. Extract CO$_2$ concentration values corresponding to observation locations and times and calculate their standard deviation.

observations

0.2  0.4  0.6  0.8  1  1.2  1.4  1.6  1.8  2   (ppm)

### $err_{f_e}$ =

1. Calculate the adjoint sensitivity of atmospheric CO$_2$ to surface CO$_2$ fluxes by region.
2. Compute the ensemble spread of the sum of posterior CO$_2$ fluxes within the 50[th] percentile adjoint sensitivity trajectory.

[$err_{f_e}$]     [Normalized adjoint sensitivity]

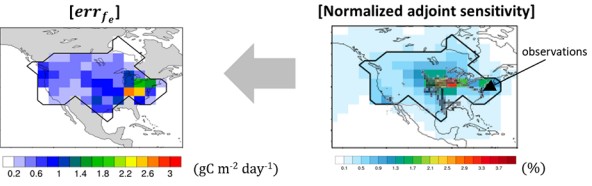

observations

0.2  0.6  1  1.4  1.8  2.2  2.6  3   (gC m$^{-2}$ day$^{-1}$)      0.1  0.5  0.8  1.3  1.7  2.1  2.5  2.9  3.3  3.7   (%)

**Figure 1.** Flow chart summarizing the process of evaluating and quantifying errors in the ensemble mean of regional posterior fluxes. RMSE$^2$ represents the mean square error between the ensemble mean of posterior CO$_2$ concentrations and observed CO$_2$ concentrations. ERR$^2_{REP}$ and ERR$^2_{OBS}$ denote estimates of representation errors and observation errors, respectively. ERR$^2_{MIP}$ represents the ensemble spread of posterior CO$_2$ concentrations. ERR$^2_{TOT}$ is defined as the sum of ERR$^2_{REP}$, ERR$^2_{OBS}$, and ERR$^2_{MIP}$, while err$_{f_e}$ and err$_{f_t}$ represent estimates of flux errors, defined as the ensemble spread of posterior fluxes, and their true values, respectively. Moreover, $h(err_{f_e})$ and $h(err_{f_t})$ denote estimates of flux errors projected onto CO$_2$ concentrations and their true values, respectively, while $\sigma^2_o$, $\sigma^2_r$, $\sigma^2_{f_t}$ ($\sigma^2_{f_e}$), $\sigma^2_t$, and $cov(\sigma_{f_t},\sigma_t)(cov(\sigma_{f_e},\sigma_t))$ refer to the types of errors represented by the error statistics – namely, observation errors, representation errors, true (estimated) flux errors projected onto CO$_2$ concentrations, transport errors, and error covariances between the preceding two terms, respectively.

were conducted at six different sites in 2017, with the majority of these observations coming from five flask measurement sites provided by the National Institute for Space Research (INPE), which likely exhibit a low bias in measured flask sample CO$_2$ mole fractions ($\sim 1$ ppm or greater) when ambient water vapor mole fractions are above $\sim 1.5\%$. These biases have been noted in some aircraft flask CO$_2$ measurements in the previous literature (Baier et al., 2020; Gatti et al., 2023), and impacted data have been removed from all other aircraft flask datasets. Despite the potential limitations of these South American observations, our analysis, aimed at introducing a method for quantifying flux errors, incorpo-

rates these data to offer guidance for future studies leveraging bias-corrected observations from this region. As discussed in more detail below, readers should keep in mind that our results from South America may exhibit lower reliability compared to those from other regions.

## 2.3 Evaluation of the ensemble posterior CO$_2$ fluxes

We first employ the two matrixes defined in Eqs. (1) and (2) to evaluate the ensemble posterior net flux errors proposed by Liu et al. (2021). One matrix is the RMSE between the ensemble mean of posterior atmospheric CO$_2$ from OCO-2

**Table 1.** Data descriptions for the airborne measurement campaigns.

| Site code | Site name | Measurement campaign name | Measurement type | Data provider | ObsPack (original) dataset identifier | Reference |
|---|---|---|---|---|---|---|
| ACG | Alaska Coast Guard, Alaska, USA | National Oceanic and Atmospheric Administration (NOAA) Global Monitoring Laboratory (GML) Aircraft Program | In situ | NOAA GML | https://doi.org/10.25925/20201204[a] | Karion et al. (2013) |
| ACT | Atmospheric Carbon and Transport – America (ACT-America), USA | ACT-America | In situ and flask | National Aeronautics and Space Administration Langley Research Center (NASA LaRC), NOAA GML | https://doi.org/10.25925/20201204[a] https://doi.org/10.3334/ORNLDAAC/1593 | Baier et al. (2020), DiGangi et al. (2021), Wei et al. (2021) |
| AirCoreNOAA | NOAA AirCore program | NOAA AirCore program | Balloon air sampler | NOAA GML | No ObsPack DOI[b] https://doi.org/10.15138/6AV0-MY81 | Karion et al. (2010) |
| ALF | Alta Floresta, Brazil | | Flask | National Institute for Space Research (INPE) | https://doi.org/10.25925/20181030[c] https://doi.org/10.1594/PANGAEA.926834 | Gatti et al. (2023) |
| CAR | Briggsdale, Colorado, USA | | Flask | NOAA GML | https://doi.org/10.25925/20210517[d] | Sweeney et al. (2015) |
| CON | Comprehensive Observation Network for TRace gases by AIrLiner (CONTRAIL) | | In situ | National Institute for Environmental Studies (NIES), Meteorological Research Institute (MRI) | https://doi.org/10.25925/20201204[a] https://doi.org/10.17595/20180208.001 | Machida et al. (2008) |
| CRV | Carbon in Arctic Reservoirs Vulnerability Experiment (CARVE), Alaska, USA | Arctic–Boreal Vulnerability Experiment (ABoVE) | In situ | NOAA GML | https://doi.org/10.25925/20201204[a] https://doi.org/10.3334/ORNLDAAC/1582 | Sweeney et al. (2022) |
| GSFC | Active Sensing of $CO_2$ Emissions over Nights, Days, and Seasons (ASCENDS), USA | ASCENDS | In situ | NASA Goddard Space Flight Center (NASA GSFC) | https://doi.org/10.25925/20201204[a] | Kawa et al. (2018) |
| IAGOS | In-service Aircraft for a Global Observing System | Civil Aircraft for the Regular Investigation of the Atmosphere Based on an Instrument Container (IAGOS-CARIBIC) | In situ | Karlsruhe Institute of Technology (IMK-ASF), Institute for Atmospheric and Environmental Sciences (IAU), Max Planck Institute for Biogeochemistry (MPI-BGC) | https://doi.org/10.25925/20201204[a] | Filges et al. (2015) |
| KORUS | The Korea–United States Air Quality (KORUS-AQ) field study | | In situ | NASA LaRC | https://doi.org/10.25925/20201204[a] https://doi.org/10.5067/ASDC/SUBORBITAL/KORUSAQ_TraceGas_AircraftInSitu_DC8_Data_1 | Vay et al. (2009) |

| Site code | Site name | Measurement campaign name | Measurement type | Data provider | ObsPack (original) dataset identifier | Reference |
|---|---|---|---|---|---|---|
| MAN | Manaus, Brazil | NOAA GML Aircraft Program | In situ | NOAA GML | https://doi.org/10.25925/20210519[e] | Stephens et al. (2018) |
| ORC | O$_2$/N$_2$ Ratio and CO$_2$ Airborne Southern Ocean (ORCAS) study | | In situ | National Center for Atmospheric Research (NCAR) | https://doi.org/10.25925/20201204[a] https://doi.org/10.5065/D6SB445X | |
| PAN | Pantanal, Mato Grosso do Sul, Brazil | | Flask | INPE | https://doi.org/10.25925/20181030[c] https://doi.org/10.25925/20181030[c] | |
| PFA | Poker Flat, Alaska, USA | NOAA GML Aircraft Program | Flask | NOAA GML | https://doi.org/10.25925/20210517[d] | Sweeney et al. (2015) |
| RBA-B | Rio Branco, Brazil | | Flask | INPE | https://doi.org/10.25925/20181030[c] https://doi.org/10.1594/PANGAEA.926834 | Gatti et al. (2023) |
| SAN | Santarém, Brazil | | Flask | INPE | https://doi.org/10.25925/20181030[c] https://doi.org/10.1594/PANGAEA.926834 | Gatti et al. (2023) |
| SGP | Southern Great Plains, Oklahoma, USA | NOAA GML Aircraft Program | Flask | The US Department of Energy (DOE) Lawrence Berkeley National Laboratory (LBNL), NOAA GML | https://doi.org/10.25925/20210517[d] | Biraud et al. (2013), Sweeney et al. (2015) |
| SONGNEX 2015 | The 2015 Shale Oil and Natural Gas Nexus air campaign, USA | The 2015 Shale Oil and Natural Gas Nexus air campaign | In situ | NOAA Chemical Sciences Laboratory (CSL) | https://doi.org/10.25925/20201204[a] | |
| TEF | Tefé, Brazil | | Flask | INPE | https://doi.org/10.25925/20181030[c] https://doi.org/10.1594/PANGAEA.926834 | Gatti et al. (2023) |
| TOM | Atmospheric Tomography Mission (ATom) | Atmospheric Tomography Mission (ATom) | In situ | NOAA GML, Harvard University | https://doi.org/10.25925/20201204[a] https://doi.org/10.3334/ORNLDAAC/1581 | Thompson et al. (2022) |

[a] obspack_co2_1_GLOBALVIEWplus_v6.1_2021-03-01 (Schuldt et al., 2021b).
[b] obspack_co2_1_AirCore_v4.0_2020-12-28.
[c] obspack_co2_1_INPE_RESTRICTED_v2.0_2018-11-13 (NOAA Carbon Cycle Group ObsPack Team, 2018).
[d] obspack_co2_1_NRT_v6.1.1_2021-05-17 (Schuldt et al., 2021a).
[e] obspack_multi-species_1_manaus_profiles_v1.0_2021-05-20 (Miller et al., 2021).

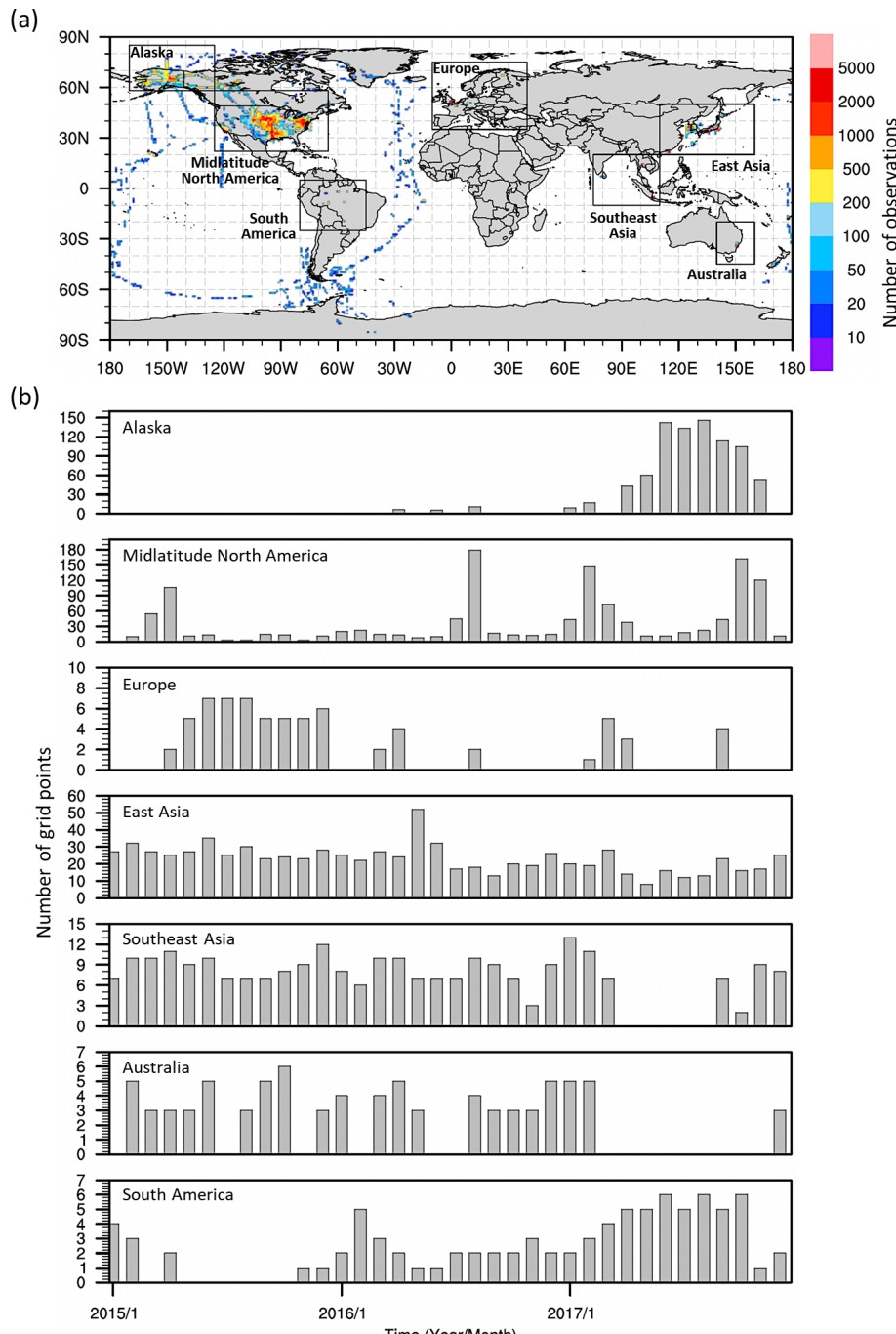

**Figure 2. (a)** The total number of airborne measurement data points used in this study at each $1° \times 1°$ grid point and **(b)** the number of $1° \times 1°$ grid points with more than 10 data points available within each region and for each month during the period 2015–2017.

MIP models and the atmospheric $CO_2$ from airborne measurements, which can be written as

$$\text{RMSE}^2 = \frac{1}{N} \sum_{i=1}^{N} \left[ \overline{h_i(\hat{x})} - y_{\text{o},i} \right] \left[ \overline{h_i(\hat{x})} - y_{\text{o},i} \right]^T ,$$

$$\text{where } \overline{h_i(\hat{x})} = \frac{1}{M} \sum_{j=1}^{M} h_{i,j}(\hat{x}_j). \quad (1)$$

Here, $\overline{h_i(\hat{x})}$ is the ensemble mean of posterior atmospheric $CO_2$ sampled at the time and location of the $i$th airborne observation, $y_{\text{o},i}$, within each $1° \times 1°$ grid cell for each month. $N$ is the monthly total number of sampled data points in each grid cell. $M$ is the number of ensemble members (i.e., 10). A single monthly RMSE value is computed using $N$ measurement data points for each grid cell. The total number

of RMSE values computed per month within each region corresponds to the number of grid cells shown in Fig. 2b. The RMSE indicates the magnitude of the actual $CO_2$ errors in the ensemble estimates, which is also a metric broadly used to evaluate the accuracy of posterior fluxes (Crowell et al., 2019; Peiro et al., 2022; Byrne et al., 2023). As illustrated in Fig. 1 and described in Appendix A (Eq. A3), $RMSE^2$ includes not only the projection of the true flux error onto $CO_2$ concentrations ($\sigma_{f_t}^2$) but also the transport error ($\sigma_t^2$), the error covariance between these two terms ($cov(\sigma_{f_t}, \sigma_t)$), the representation error ($\sigma_r^2$), and the airborne observation error ($\sigma_o^2$). Both transport errors and representation errors stem from transport models. Transport errors include errors in model structures and meteorological fields, while representation errors arise from a mismatch in resolution between model simulations and observations.

In practice, true flux errors are often approximated by the spread of ensemble fluxes, so the sum of projection of true flux errors to $CO_2$ concentrations and transport errors is approximated by the ensemble spread of simulated $CO_2$ concentrations in the OCO-2 MIP, as shown in Appendix A. To evaluate whether this approximation represents the true errors in the ensemble mean fluxes and mean simulated $CO_2$ concentrations, we define another quantity, $ERR_{TOT}^2$ (Fig. 1). Unlike the RMSE, the variance terms of the flux error ($\sigma_{f_e}^2$) and transport error ($\sigma_t^2$), as well as the covariance term between these errors ($cov(\sigma_{f_e}, \sigma_t)$), are replaced by the spread of the ensemble (i.e., variance) of posterior atmospheric $CO_2$ concentrations ($ERR_{MIP}^2$), defined as

$$ERR_{MIP}^2 = \frac{1}{N} \sum_{i=1}^{N} \frac{1}{M} \sum_{j=1}^{M} \left[ h_{i,j}(\hat{x}_j) - \overline{h_i(\hat{x})} \right] \left[ h_{i,j}(\hat{x}_j) - \overline{h_i(\hat{x})} \right]^T. \tag{2}$$

Unlike in Liu et al. (2021), which used only one transport model, $ERR_{MIP}^2$ accounts for transport errors because posterior atmospheric $CO_2$ was generated by multiple types of transport models in the OCO-2 MIP, driven by different meteorology fields. Thus, the $ERR_{MIP}^2$ term accounts for transport errors but not representation errors, due to the coarse spatial resolution of these transport models, with the highest spatial resolution being $2° \times 2.5°$.

To obtain representation errors and observation errors not captured by $ERR_{MIP}^2$, we additionally calculate $ERR_{REP}^2$ and $ERR_{obs}^2$, respectively. $ERR_{REP}^2$ denotes the representation error ($\sigma_r^2$) in $RMSE^2$, as shown in Fig. 1. It is defined as the spatial variability in atmospheric $CO_2$ within a $2° \times 2.5°$ grid cell and is written as

$$ERR_{REP}^2 = \frac{1}{N} \sum_{i=1}^{N} VAR_{CO_2, i}. \tag{3}$$

Using high-resolution ($0.5° \times 0.625°$) 3-hourly GEOS-5 simulation results for 2018 from the NASA Goddard Space Flight Center (Weir et al., 2021), we calculate the variance of

atmospheric $CO_2$ concentrations within each $2° \times 2.5°$ grid cell at each 3 h interval. Then, we sample the $CO_2$ variance ($VAR_{CO_2, i}$) value corresponding to the grid cell containing the $i$th observation and at the time closest to that of the observation. Subsequently, the monthly mean value of the $N$ co-sampled variances is derived ($ERR_{REP}^2$). We assume that the variances do not vary significantly across years, given the lower monthly variability in $ERR_{REP}$ compared to that in the RMSE and $ERR_{MIP}$ (as will be shown in Sect. 3.2). The reason for calculating the $CO_2$ variance within a $2° \times 2.5°$ grid is that this grid represents the finest resolution among the OCO-2 MIP models. We evaluate whether the representation errors derived from simulated atmospheric $CO_2$ fields represent the actual spatial variability in $CO_2$ concentrations by comparing the simulated $CO_2$ variance with the spatial variance of airborne measurement data from the ACT-America project (see the Supplement and Fig. S1). The evaluation results support our approach.

$ERR_{OBS}^2$ represents the observation error ($\sigma_o^2$) in $RMSE^2$, as shown in Fig. 1. Unfortunately, this information is missing from many of the airborne measurement datasets included in the given OCO-2 MIP ObsPack format, even though uncertainties may be included in the original datasets. The World Meteorological Organization (WMO) community has established network compatibility objectives for the precision of atmospheric $CO_2$ measurements: 0.1 ppm for the Northern Hemisphere and 0.05 ppm for the Southern Hemisphere. Assuming an ideal situation without any systematic bias, we set the observation error ($ERR_{OBS}$) for all airborne observations to 0.1 ppm. However, in reality, systematic errors could be present in airborne observations, stemming from instrument or setup biases, calibration offsets, and other factors. In particular, $CO_2$ measurements from INPE taken in South America might exhibit higher measurement errors compared to those taken in other regions due to unresolved water vapor contamination issues in the flask measurements, which may result in both a low bias ($\sim 1$–3 ppm at 3 % absolute humidity) and spurious variability (Baier et al., 2020). The potential effects of these systematic errors on our findings will be addressed in Sect. 4. This study employs only $ERR_{OBS}^2$ for calculating $ERR_{TOT}^2$ and does not compare it with other error quantities from Sect. 3.

Therefore, $ERR_{TOT}$, the approximation for the RMSE, is defined as

$$ERR_{TOT}^2 = ERR_{OBS}^2 + ERR_{REP}^2 + ERR_{MIP}^2. \tag{4}$$

By applying 1000 bootstrap resamplings to the monthly grid-based error statistics (e.g., the RMSE, $ERR_{MIP}$, $ERR_{REP}$, and $ERR_{TOT}$) within each region, we obtain regional mean values for these error statistics, along with their corresponding 95 % confidence intervals.

To evaluate whether the spread of ensemble $CO_2$ fluxes from the OCO-2 MIP represents the true flux errors in the ensemble mean, we calculate the ratio between monthly

$ERR_{TOT}$ and RMSE values as follows:

$$\text{Ratio}^2 = \frac{ERR_{TOT}^2}{RMSE^2}. \tag{5}$$

Given that $ERR_{REP}^2$ reasonably depicts actual representation errors, $\text{Ratio}^2$ can indicate whether posterior flux and transport errors computed from the ensemble spread overestimate or underestimate the true flux and transport errors. In this study, we assume that the estimated transport errors from the ensemble spread among the transport models used in the OCO-2 MIP represent the true transport errors and that the difference between $RMSE^2$ and $ERR_{TOT}^2$ mainly arises from the difference in the flux error variances ($\sigma_{f_t}^2$ and $\sigma_{f_e}^2$). Thus, a ratio close to 1 indicates that the estimated posterior flux errors derived from the ensemble model spread are close to the true posterior flux errors in the ensemble mean fluxes. A ratio greater than 1 means that the posterior flux errors are overestimated and vice versa. However, our assumption regarding the transport errors may be a strong one given that they are derived from 10 ensemble members covering four different transport models, which might not fully capture the actual transport errors. We discuss how this assumption affects our key results in Sect. 4.

## 2.4 Quantification of uncertainties in the ensemble mean of the posterior $CO_2$ fluxes

In addition to qualitative evaluations of posterior flux errors using the ratios between $ERR_{TOT}$ and the RMSE, we propose a method for quantitatively assessing the ensemble posterior flux errors (i.e., the variance of flux errors) in both $CO_2$ space and flux space. To do this, we first need to calculate the variance of atmospheric $CO_2$ errors due solely to the ensemble spread of posterior fluxes from the OCO-2 MIP ($h\left(\text{err}_{f_e}\right)^2$). As shown in Appendix A, this term can be written as

$$h\left(\text{err}_{f_e}\right)^2 = \frac{1}{N}\sum_{i=1}^{N}\frac{1}{M}\sum_{k=1}^{M}\frac{1}{M}\sum_{j=1}^{M} \\ \left[h_k\left(\hat{x}_{k,i}\right) - h_k\left(\hat{x}_{j,i}\right)\right]\left[h_k\left(\hat{x}_{k,i}\right) - h_k\left(\hat{x}_{j,i}\right)\right]^T. \tag{6}$$

All transport models engaged in the OCO-2 MIP would ideally be used to derive $h\left(\text{err}_{f_e}\right)^2$. However, in this study, we approximate this error term using the GEOS-Chem model, as depicted in the following:

$$h\left(\text{err}_{f_e}\right)^2 \approx h_{GC}\left(\text{err}_{f_e}\right)^2 = \frac{1}{N}\sum_{i=1}^{N}\frac{1}{M}\sum_{j=1}^{M} \\ \left[\overline{h_{GC}\left(\hat{x}_i\right)} - h_{GC}\left(\hat{x}_{j,i}\right)\right]\left[\overline{h_{GC}\left(\hat{x}_i\right)} - h_{GC}\left(\hat{x}_{j,i}\right)\right]^T, \tag{7}$$

where $\overline{h_{GC}\left(\hat{x}_i\right)} = \frac{1}{M}\sum_{j=1}^{M}h_{GC}\left(\hat{x}_{j,i}\right)$.

To obtain $h_{GC}\left(\text{err}_{f_e}\right)^2$, we conduct a set of forward simulations using the GEOS-Chem transport model (within the GEOS-Chem adjoint model (v35j); Henze et al., 2007).

In all 10 experiments, consistent meteorology and emission forcing data from version 2 of the Modern-Era Retrospective analysis for Research and Applications (MERRA-2; Gelaro et al., 2017) and the Open-source Data Inventory for Anthropogenic $CO_2$ (ODIAC; Oda and Maksyutov, 2015) are used. Identical monthly balanced, hourly terrestrial-biosphere fluxes from SiB4 (Haynes et al., 2021) are also employed. However, in each experiment, the prescribed monthly fluxes of terrestrial ecosystems and oceans are based on posterior fluxes from the respective 10 OCO-2 MIP ensemble members. All experiments are performed at a $2° \times 2.5°$ horizontal resolution and with 47 vertical levels for the period 2015–2017. By calculating the mean of the variances of simulated $CO_2$ concentrations among the 10 experiments for the $i$th airborne observation within each $1° \times 1°$ grid cell, we derive $h_{GC}\left(\text{err}_{f_e}\right)^2$.

Because we assume that the spread of ensemble transport models used in the OCO-2 MIP represents the true transport errors included in $RMSE^2$, the transport errors, along with the observation errors and representation errors, cancel out when we calculate the difference between monthly $RMSE^2$ and $ERR_{TOT}^2$ values. Consequently, the difference between monthly $RMSE^2$ and $ERR_{TOT}^2$ values arises from the differences in the flux error variances ($\sigma_{f_t}^2$ and $\sigma_{f_e}^2$). The difference between the monthly true flux error ($h\left(\text{err}_{f_t}\right)^2$) and the estimated flux error ($h\left(\text{err}_{f_e}\right)^2 \approx h_{GC}\left(\text{err}_{f_e}\right)^2$) projected onto $CO_2$ space can be derived from the difference between $RMSE^2$ and $ERR_{TOT}^2$ as follows:

$$h\left(\text{err}_{f_t}\right)^2 - h\left(\text{err}_{f_e}\right)^2 = RMSE^2 - ERR_{TOT}^2. \tag{8}$$

From Eq. (8), we can derive the true error in the ensemble mean fluxes in $CO_2$ space, $h\left(\text{err}_{f_t}\right)^2$. In 158 out of 181 cases, representing the total number of observation months across all seven regions, $h\left(\text{err}_{f_t}\right)$ can be derived using this equation. In 23 cases (13 % of the cases), $h\left(\text{err}_{f_t}\right)$ cannot be derived when $ERR_{TOT}$ and/or $h\left(\text{err}_{f_e}\right)$ values fall outside the applicable range. Around 40 % of the exception cases occur in South America, where observations cover only one to six $1° \times 1°$ grid cells per month, suggesting that observations are insufficient for quantifying the monthly flux errors in this region.

In order to link these terms with flux errors in flux space, we first identify the areas sensitive to airborne $CO_2$ measurements by conducting sensitivity experiments using the GEOS-Chem adjoint model. Seven sets of adjoint sensitivity experiments are conducted to examine the sensitivity of airborne measurements in each region (as defined in Fig. 2a) to surface $CO_2$ fluxes for the month of observations. The sensitivity experiments use the same meteorology and $CO_2$ emission datasets as the forward simulations, along with the ensemble mean of posterior terrestrial-biosphere flux and ocean flux values. The following explanation of the sensitivity analysis uses the same notation as Liu et al. (2015). The cost

function ($J$) is defined as the sum of the simulated $CO_2$ concentrations at the locations of airborne observations within each region and for each month:

$$J = \sum_{i=1}^{N} h_i(\hat{x}). \tag{9}$$

The sensitivity of observations to surface fluxes at the $l$th grid cell and $t$th time is derived from the partial derivative of $J$ with respect to surface fluxes ($\hat{x}_{l,k}$), expressed as

$$\gamma_{l,t} = \frac{\partial J}{\partial \hat{x}_{l,t}}. \tag{10}$$

The monthly cumulative sensitivity ($\beta$) with respect to surface fluxes is determined by integrating $\gamma_{l,t}$ from the measurement time ($t_0$) to the initial time ($t_{-T}$) for each month:

$$\beta_l = \sum_{t=t_0}^{t-T} \gamma_{l,t}. \tag{11}$$

In order to identify the areas most sensitive to the airborne observations, we select the areas accounting for 50 % of the global total values of $\beta$ for each region and month. Areas with sensitivity values lower than 0.1 % (0.15 % for Alaska, Australia, and Southeast Asia) of the total value of $\beta$ are excluded due to occasional cases where observations are uniformly influenced across excessively wide regions as a result of active atmospheric mixing. Additionally, to avoid the excessive consideration of localized effects caused by a large number of observations occurring in a single location, regions with sensitivity values greater than 1 % are included in the effective area. We then compute the estimated posterior flux error in flux space ($\text{err}_{f_e}^2 = \sigma_{f_e}^2$) by calculating the ensemble spread of the total posterior flux values (and area-averaged mean values) over the effective area for each month during the period 2015–2017, as illustrated in Fig. 1. The estimated mean posterior flux error ($\text{err}_{f_e}$) over the selected areas for each month exhibits a significant correlation ($p \leq 0.05$) with monthly $h\left(\text{err}_{f_e}\right)$ values across all regions, except for Australia, where the observational campaign was conducted during specific months (Fig. S2). While the observed atmospheric $CO_2$ concentration is influenced by both land and ocean sources, a comparison of the magnitudes of $\text{err}_{f_e}$ between ocean and land within the effective areas reveals that, on average, land flux errors contribute more than 95 % to the total flux errors in all regions (Fig. S3). This result indicates that our evaluation results, based on atmospheric $CO_2$, can be applied to deriving the actual errors in posterior net land $CO_2$ fluxes within the selected areas in flux space.

This study provides both monthly and 3-year mean values of regional flux error statistics for the period 2015–2017. Technically, it is possible to derive monthly true errors in the ensemble mean of net land $CO_2$ fluxes using monthly error statistics. However, to obtain more robust results, we compute the true errors in annual total fluxes over the analysis period. To identify the areas contributing most to the computed

mean error statistics, we calculate the number of months selected as effective areas for monthly airborne observations. These grid cells, at a $2° \times 2.5°$ resolution (corresponding to the effective areas), are assigned a value of 1, while the remaining cells are assigned a value of 0 for each month. We then calculate composite values for each grid cell over the 3 years. A higher number of months indicates that more information from those grid cells is utilized in calculating the 3-year regional mean error statistics. We define our 3-year mean error statistics as mostly representing the areas where the composite values exceed 8, corresponding to 20 % of the total months analyzed (i.e., 36 months).

The observation operator, which converts surface $CO_2$ fluxes to atmospheric $CO_2$, is generally assumed to be linear. Therefore, in these areas, we can obtain the true error in the ensemble annual total net land fluxes, $\text{err}_{f_t} \left(= \sigma_{f_t}\right)$, by multiplying the ratio between the 3-year mean values of $h\left(\text{err}_{f_t}\right)$ and $h\left(\text{err}_{f_e}\right)$ by the ensemble spread of the annual total net land flux estimates ($\text{err}_{f_e}$) within the effective areas. The equation can be written as

$$\text{err}_{f_t} = \frac{h\left(\text{err}_{f_t}\right)}{h\left(\text{err}_{f_e}\right)} \times \text{err}_{f_e}. \tag{12}$$

One thing readers should keep in mind is that $\text{err}_{f_e}$ is identical to the ensemble spread of posterior terrestrial-biosphere fluxes because all OCO-2 MIP models used uniform fossil fuel emission estimates and assumed them to be perfectly known. Lastly, to explore the characteristics of regions where the average annual total $\text{err}_{f_t}$ value is significantly underestimated, we compute the ensemble mean of the average annual posterior terrestrial-biosphere $CO_2$ fluxes and the fossil fuel $CO_2$ emissions (from the ODIAC dataset) for the effective area.

## 3   Results

### 3.1   Spatiotemporal variations in ensemble posterior $CO_2$ concentration errors and other major error components

Because the magnitude of land–atmosphere $CO_2$ fluxes is generally over 10 times greater than that of ocean–atmosphere $CO_2$ fluxes, the observed atmospheric $CO_2$ over the oceans carries signals from nearby land fluxes. The four ATom campaigns, spanning four seasons, and the ORCAS campaign, conducted during austral summer, covered wide latitudinal ranges, primarily over the oceans, providing a unique opportunity for analyzing the latitudinal distributions of inverse-modeling errors and the contributions of major error sources. We compare the ensemble posterior $CO_2$ to airborne $CO_2$ measurements taken between 1 and 5 km a.g.l. and then calculate the mean error statistics for the entire campaign period. Comparisons with observations from the ATom and ORCAS campaigns reveal a general increase in RMSE

values toward the northern high latitudes, reaching 1.2 ppm at 40° N (Fig. 3a and f). The latitudinal gradient becomes particularly evident during the summer season, with RMSE values exceeding 1.5 ppm over North America (Fig. S4), suggesting significant contributions of the errors in land fluxes to the differences between observed and simulated atmospheric CO$_2$. Additionally, consistently elevated RMSE values ($> 1.5$ ppm) commonly appear over the west coast of Africa throughout the seasons.

Both ERR$_{MIP}$ and ERR$_{REP}$ exhibit spatial distributions similar to that of the RMSE (Fig. 3a–c and f). However, ERR$_{MIP}$ has a stronger positive correlation with the RMSE ($r = 0.57$ and 0.58 for the ATom and ORCAS campaigns, respectively) than ERR$_{REP}$ ($r = 0.35$ and 0.32), with an average magnitude (0.49 and 0.32 ppm) greater than that of ERR$_{REP}$ (0.27 and 0.20 ppm) globally across both campaign periods. In particular, ERR$_{MIP}$ (ERR$_{REP}$) accounts for 75 % (37 %) of the anomalously high RMSE values (1.5 ppm) in North America (32–50° N and 85–124° W) and 75 % (30 %) of the RMSE values (1.2 ppm) along the west coast of Africa. These findings indicate that ERR$_{MIP}$, representing errors in posterior fluxes and transport, is the most significant factor in explaining the RMSE.

Next, in order to assess the proximity of the estimated posterior flux errors, based on the spread of the OCO-2 MIP ensemble fluxes, to the true posterior flux errors in the ensemble mean, we compare the RMSE with the sum of ERR$_{MIP}$, ERR$_{REP}$, and ERR$_{OBS}$ (referred to as ERR$_{TOT}$). The ratio of ERR$_{TOT}$ to the RMSE exceeds 1 over the tropical Pacific and the Southern Ocean (Fig. 3d and e), indicating that the ensemble spread of posterior fluxes overestimates true flux errors over regions sensitive to these observations. This overestimation pattern consistently appears for both the ATom and ORCAS campaigns across all seasons (Fig. S5). Airborne CO$_2$ measurements for this area are predominantly influenced by ocean fluxes due to the limited land extent and the significant distance from land (Yun et al., 2022), suggesting that the true posterior ocean flux errors may be smaller than the spread of the ensemble posterior flux estimates. In contrast, a ratio between ERR$_{TOT}$ and the RMSE of less than 1 was observed along the African coast during the ATom campaigns, with the exception of the 2018 spring campaign, which was conducted in a region relatively far from Africa. Considering that these airborne observations are known to be sensitive to terrestrial-biosphere fluxes in tropical Africa (Liu et al., 2021), our results imply that the true errors in the ensemble mean terrestrial-biosphere fluxes in this region may be larger than the estimated errors based on the OCO-2 MIP ensemble spread. These findings agree with Gaubert et al. (2023), who show that most inverse models assimilating OCO-2 XCO$_2$ retrievals tend to overestimate the net carbon sources in this region due to potential positive biases in the OCO-2 retrievals.

In the northern mid-to-high latitudes, characterized by significant land CO$_2$ flux impacts on atmospheric CO$_2$ varia-

tions (Yun et al., 2022), the ratio between ERR$_{TOT}$ and the RMSE exhibits substantial variation across space and time. The ratio between ERR$_{TOT}$ and the RMSE is greater than 1 within the North American continent during summer and fall. However, in other areas, there is a mixed pattern involving ratios both below and above 1, although the majority of areas exhibit ratios less than 1 during winter. These findings highlight that the degree of underestimation or overestimation of true flux errors, based on the ensemble spread, can vary depending on the region and season, emphasizing the need for a more detailed evaluation of flux errors at the regional level using long-term independent observations.

## 3.2 Evaluation of OCO-2 v10 MIP ensemble posterior CO$_2$ flux errors by region

In this section, we calculate regionally averaged monthly error statistics by comparing the ensemble posterior CO$_2$ to airborne measurements taken over seven regions during 2015–2017. RMSE values across all these regions exhibit significant monthly variations, with values falling within the range of 1–3 ppm and no clear seasonality, possibly due to variations in observation routes (Fig. 4). Consistent with the results shown in Sect. 3.1, ERR$_{MIP}$ is the most significant factor explaining the variations in the RMSE. Among the seven regions, significant positive correlations ($p < 0.05$) between monthly RMSE and ERR$_{MIP}$ values are observed in Alaska ($r = 0.46$), midlatitude North America ($r = 0.63$), Europe ($r = 0.60$), and East Asia ($r = 0.60$). Furthermore, the correlation coefficient is greater than or comparable to that concerning ERR$_{REP}$. This suggests that, in these regions, temporal variations in the errors in posterior fluxes and transport are the major contributors to temporal variations in the RMSE. On the other hand, the RMSE does not exhibit a significant correlation with either ERR$_{MIP}$ or ERR$_{REP}$ in Southeast Asia, Australia, and South America. This implies that the estimated posterior flux errors based on the ensemble spread may not represent the temporal variations in true flux errors in these regions.

RMSE values exhibit significant variability not only over time but also across regions. The 3-year average RMSE is highest in East Asia (mean: 1.98 ppm; 95 % confidence intervals: [1.90, 2.06] ppm), followed by Europe (1.57, [1.41, 1.74] ppm), and lowest in Australia (0.88, [0.79, 0.97] ppm), with Alaska having a slightly higher RMSE (1.19, [1.12, 1.25] ppm). ERR$_{MIP}$ is the primary error component of the RMSE, accounting for 58 %–83 % of the RMSE and surpassing the ERR$_{REP}$ values for all the regions by a factor of 1.2–2.1. In East Asia, the difference between ERR$_{MIP}$ and ERR$_{REP}$ is relatively small compared to that observed in other regions. This could be attributed to the presence of numerous significant carbon sources, particularly along coastal areas, resulting in increased spatial variability in CO$_2$ within the coarse grid cells of the OCO-2 MIP inverse modeling.

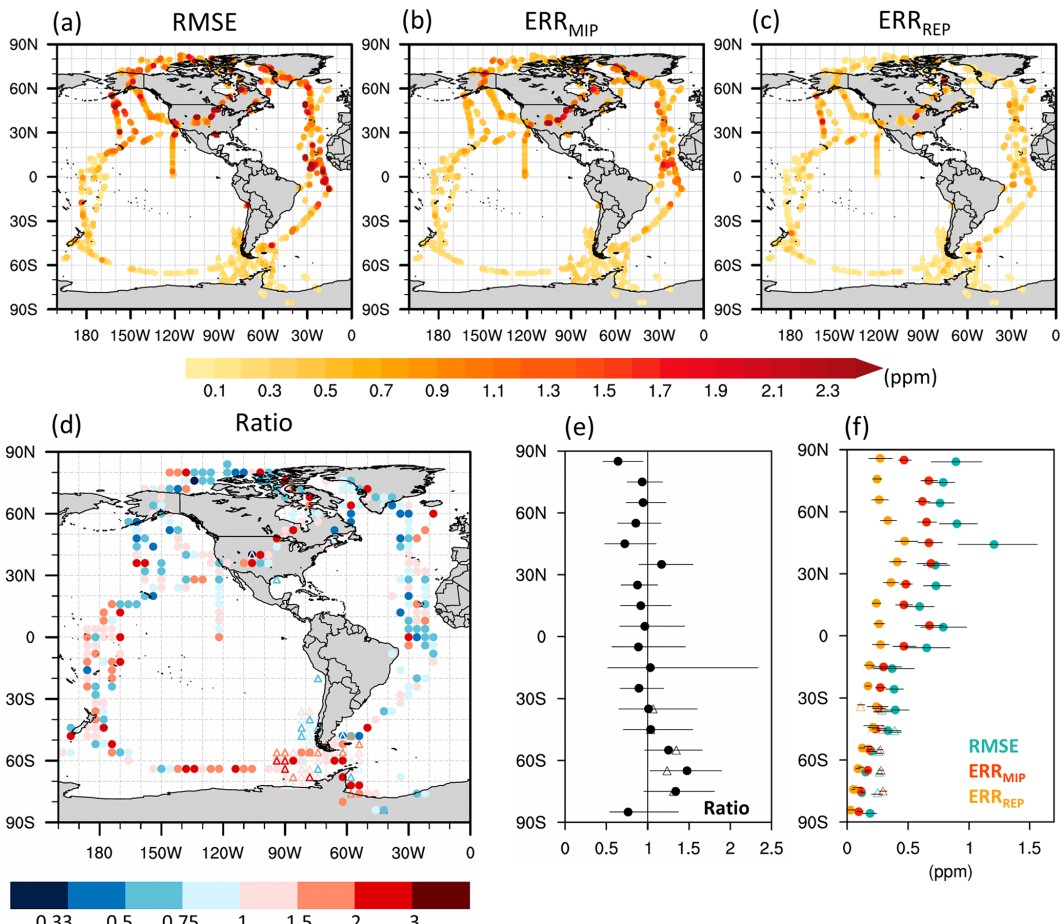

**Figure 3.** Spatial distributions of **(a)** RMSE, **(b)** $\mathrm{ERR_{MIP}}$, **(c)** $\mathrm{ERR_{REP}}$, and **(d)** Ratio $\left( \sqrt{(\mathrm{ERR_{OBS}})^2 + \mathrm{ERR^2_{REP}} + \mathrm{ERR^2_{MIP}}} / \mathrm{RMSE} \right)$ values with respect to where ATom airborne measurements (circles) and ORCAS airborne measurements (triangles) were taken and **(e, f)** their latitudinal distributions, smoothed by a 10° moving average with 95 % confidence intervals derived from 1000 bootstrap samples of datasets (error bars).

The ratio between $\mathrm{ERR_{TOT}}$ and the RMSE also shows significant variability across regions. Our results indicate that, on average, the estimated flux errors for Alaska and South America closely match the true flux errors, with ratios of 0.98 [0.89, 1.08] and 0.99 [0.79, 1.24], respectively, while mid-latitude North America, Europe, East Asia, Southeast Asia, and Australia show significant underestimations at the 95 % confidence level, with ratios of 0.90 [0.83, 0.97], 0.79 [0.61, 0.97], 0.84 [0.78, 0.91], 0.75 [0.65, 0.86], and 0.73 [0.59, 0.87], respectively, throughout the analysis period. Furthermore, the monthly variability (i.e., standard deviation) of the ratios is much greater in regions with diverse campaign durations and routes, such as South America (0.87), than in East Asia (0.21), characterized by a consistent 3-year observation campaign along the same paths. This suggests that the spatial variability in the degree of flux error underestimation or overestimation may exceed the temporal variability.

### 3.3 Error quantification of OCO-2 v10 MIP ensemble posterior net CO₂ fluxes by region

Next, by incorporating the monthly RMSE, $\mathrm{ERR_{TOT}}$, and $h\left(\mathrm{err_{f_e}}\right)$ values, we derive the monthly true posterior flux error in CO₂ space (i.e., $h\left(\mathrm{err_{f_t}}\right)$) for each region with respect to the period 2015–2017 (Fig. 5). Regionally averaged $h\left(\mathrm{err_{f_t}}\right)$ values exhibit different seasonal and monthly variability compared to $h\left(\mathrm{err_{f_e}}\right)$ values. In the northern midlatitude regions, $h\left(\mathrm{err_{f_t}}\right)$ shows clear seasonal cycles for the entire analysis period, despite there being different observation routes for each month. For example, in midlatitude North America and East Asia, the growing season (May to October) exhibits higher $h\left(\mathrm{err_{f_e}}\right)$ values (0.6 and 0.9 ppm, respectively) than the non-growing season (November to April; 0.4 and 0.7 ppm, respectively). Seasonal variations in $h\left(\mathrm{err_{f_t}}\right)$ are also observed in East Asia and partially observed in midlatitude North America with respect to 2017, but they are not discernible in Alaska and Europe. In addi-

tion, monthly $h\left(\mathrm{err}_{\mathrm{f_t}}\right)$ values do not exhibit a significant correlation ($p < 0.05$) with monthly $h\left(\mathrm{err}_{\mathrm{f_e}}\right)$ values in Alaska, Southeast Asia, and South America. Moreover, $h\left(\mathrm{err}_{\mathrm{f_t}}\right)$ displays greater monthly variability than $h\left(\mathrm{err}_{\mathrm{f_e}}\right)$. For example, in midlatitude North America and East Asia, the standard deviation of monthly $h\left(\mathrm{err}_{\mathrm{f_t}}\right)$ values is 1.8 and 2.3 times greater than that of monthly $h\left(\mathrm{err}_{\mathrm{f_e}}\right)$ values, respectively.

The comparison between the 3-year average $h\left(\mathrm{err}_{\mathrm{f_t}}\right)$ value and the RMSE highlights the substantial contributions of posterior flux errors to the differences between airborne observations and simulated atmospheric $CO_2$ from OCO-2 MIP ensemble models. The $h\left(\mathrm{err}_{\mathrm{f_t}}\right)$ value tends to be larger in regions with a higher RMSE, peaking in East Asia ($h\left(\mathrm{err}_{\mathrm{f_t}}\right) = 1.32$ ppm; RMSE $= 1.98$ ppm) and reaching a minimum in Australia ($h\left(\mathrm{err}_{\mathrm{f_t}}\right) = 0.75$ ppm; RMSE $= 0.88$ ppm) (Figs. 4h and 5h). The $h\left(\mathrm{err}_{\mathrm{f_t}}\right)$ value accounts for up to 85 % of the RMSE in Australia, followed by Southeast Asia (80 %), and accounts for a minimum of 60 % of the RMSE in South America, with the contribution in midlatitude North America being slightly higher (64 %). This indicates the dominant contributions of posterior flux errors to the RMSE, surpassing the contributions of representation and transport errors in the first two regions.

Throughout the analysis period, the regional mean ratios between $h\left(\mathrm{err}_{\mathrm{f_e}}\right)$ and $h\left(\mathrm{err}_{\mathrm{f_t}}\right)$ indicate significant underestimations, at a 95 % confidence level, of true posterior flux errors in midlatitude North America, Europe, East Asia, Southeast Asia, and Australia by factors of 0.74 [0.61, 0.88], 0.52 [0.27, 0.78], 0.59 [0.48, 0.70], 0.56 [0.41, 0.72], and 0.59 [0.34, 0.87], respectively (Fig. 5h). In contrast, Alaska and South America exhibit comparable estimates of true flux errors, with factors of 0.96 [0.76, 1.17] and 0.97 [0.49, 1.54], respectively. The regions with significant underestimations align with those identified in the previous analysis based on the ratios between $\mathrm{ERR_{TOT}}$ and the RMSE (Sect. 3.2), but the ratios between $h\left(\mathrm{err}_{\mathrm{f_e}}\right)$ and $h\left(\mathrm{err}_{\mathrm{f_t}}\right)$ imply a stronger underestimation of the true flux errors. The ratios have larger uncertainty ranges in regions where observations were conducted over limited durations and locations, such as Europe, Australia, and South America, than in midlatitude North America and East Asia, where observations cover wider areas and occur more frequently.

Finally, using the 3-year regional mean ratios between $h\left(\mathrm{err}_{\mathrm{f_e}}\right)$ and $h\left(\mathrm{err}_{\mathrm{f_t}}\right)$, we compute the true errors in the annual net land fluxes over the effective areas, averaged for the period 2015–2017 (Fig. 6). We find that the actual flux errors are underestimated, particularly in regions where annual fossil fuel $CO_2$ emissions exceed annual terrestrial-biosphere fluxes by a factor of 3–31. The airborne measurements carried out in midlatitude North America, East Asia, and Southeast Asia are influenced by a broad region encompassing the United States, the eastern part of East Asia, and the western part of Southeast Asia, where fossil fuel $CO_2$ emissions correspond to values of 1341, 2443, and 815 $\mathrm{Tg\,C\,yr^{-1}}$, respectively. The first two areas

are estimated as significant terrestrial-biosphere $CO_2$ sinks, with estimated fluxes of $-414 \pm 279$ (ensemble mean $\pm 1\sigma$) and $-561 \pm 380\,\mathrm{Tg\,C\,yr^{-1}}$, in contrast to Southeast Asia ($26 \pm 118\,\mathrm{Tg\,C\,yr^{-1}}$). However, the $CO_2$ sinks are more than 3 and 4 times smaller than the fossil fuel $CO_2$ emissions, respectively. The recalculated net land flux errors in these regions exceed the ensemble spread, with values of 374, 643, and 211 $\mathrm{Tg\,C\,yr^{-1}}$, respectively. Observations across Europe and Australia, conducted over limited periods and at specific locations, mainly represent certain areas in western Europe and the southeastern part of Australia, where fossil fuel emissions (234 and 53 $\mathrm{Tg\,C\,yr^{-1}}$, respectively) are around 4 and 5 times greater than terrestrial-biosphere sinks ($-51 \pm 34$ and $-10 \pm 67\,\mathrm{Tg\,C\,yr^{-1}}$, respectively). The recalculated net land flux errors in these regions are also larger than the ensemble spread, with estimates of 65 and 114 $\mathrm{Tg\,C\,yr^{-1}}$, respectively. In contrast, the most influential areas for observations in Alaska and South America, encompassing the southeastern region of Alaska and the northern part of Brazil, are characterized as terrestrial-biosphere sinks of $-8 \pm 11\,\mathrm{Tg\,C\,yr^{-1}}$ and sources of $625 \pm 387\,\mathrm{Tg\,C\,yr^{-1}}$, respectively. These values are comparable to, or more than 10 times greater than, those of fossil fuel emissions (10 and 38 $\mathrm{Tg\,C\,yr^{-1}}$, respectively). The observation-based estimates of true net land flux errors are almost identical to the ensemble spread in both regions, with values of 11 and 398 $\mathrm{Tg\,C\,yr^{-1}}$, respectively.

## 4 Discussion and conclusions

Our results show that errors in posterior net land $CO_2$ fluxes are a major contributor to the RMSE between simulated posterior $CO_2$ and airborne observations for the period 2015–2017. Our findings reaffirm the feasibility of evaluating inversion performance on land flux estimates through a direct comparison between airborne observations and model data (Houweling et al., 2015; Chevallier et al., 2019; Crowell et al., 2019; Byrne et al., 2023). However, when evaluating inversion estimates at regional scales, the significance of representation and transport errors becomes pronounced. Our results show that regional variations in representation errors, along with the sum of transport errors and their covariances with flux errors (inferred from the difference between $\mathrm{ERR_{MIP}}$ and $h\left(\mathrm{err}_{\mathrm{f_e}}\right)$; Fig. S6), exceed those in true flux errors projected onto $CO_2$ space, indicating that regional differences in the RMSE do not directly correspond to differences in flux errors. For example, although the 3-year mean errors in representation and transport for East Asia exceed those for Southeast Asia by 0.5 and 0.3 ppm, respectively, the disparity in the mean true flux errors projected onto $CO_2$ space between the two regions is only 0.2 ppm. This result is supported by previous studies, highlighting that spatial distributions of simulated $CO_2$ concentrations can vary significantly depending on the transport model (Schuh et al., 2019) and its spatial resolution (Stanevich et al., 2020). Therefore,

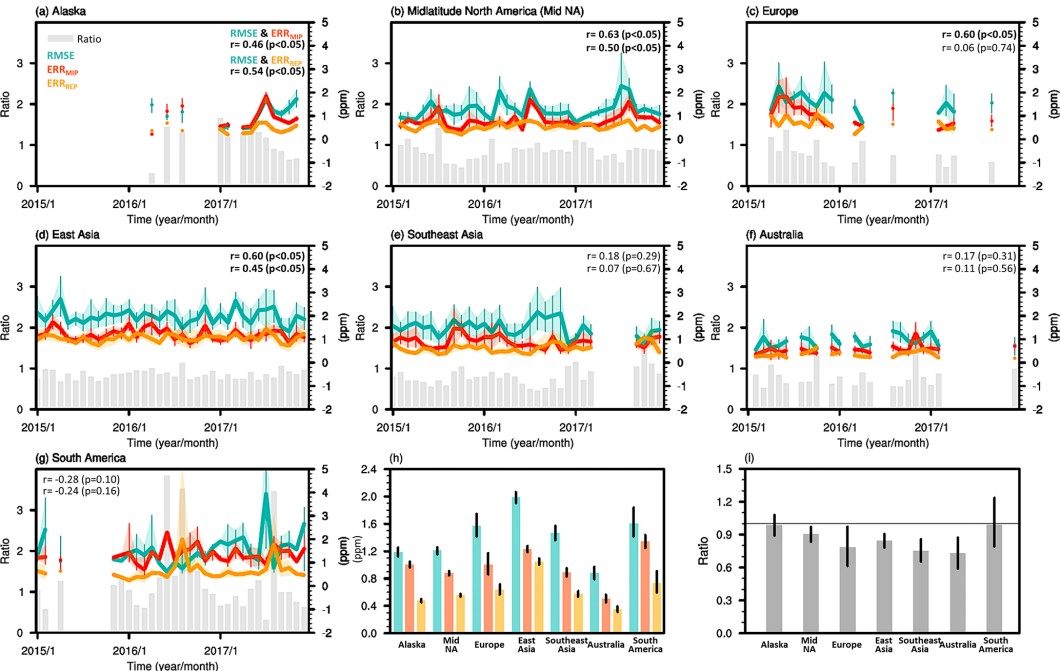

**Figure 4. (a–g)** Monthly RMSE, ERR$_{MIP}$, ERR$_{REP}$, and Ratio values for each region and **(h, i)** the corresponding mean values for the period 2015–2017. The upper-right numbers in panels **(a)**–**(g)** indicate the correlation coefficient between the RMSE and ERR$_{MIP}$ and that between the RMSE and ERR$_{REP}$. The shaded areas and error bars represent 95 % confidence intervals derived from 1000 bootstrap samples of datasets.

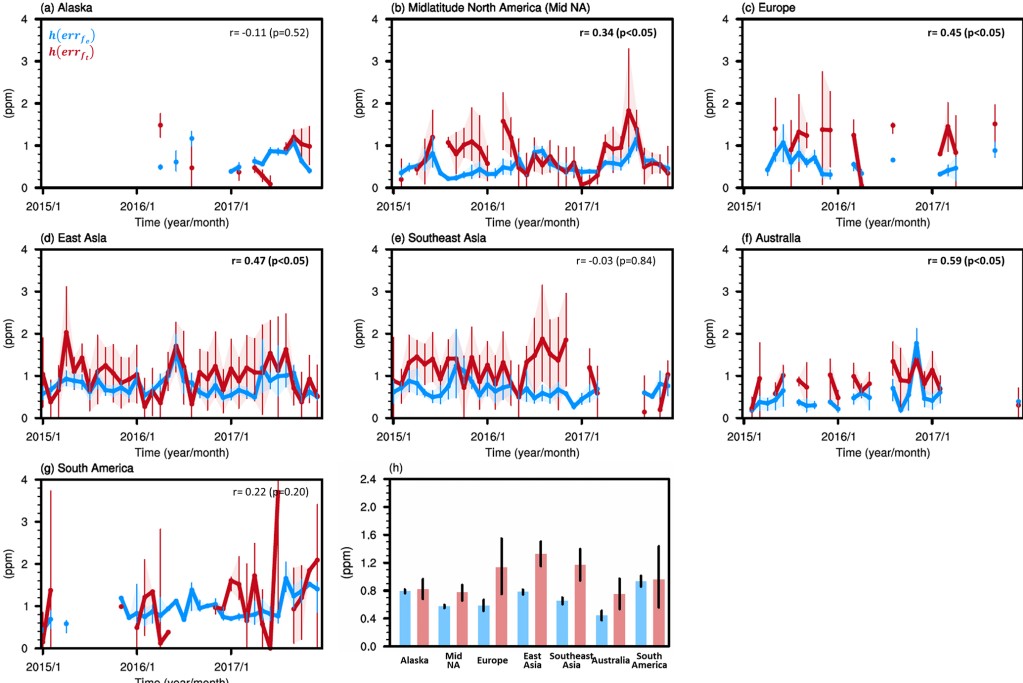

**Figure 5. (a–g)** Monthly values of $h\left(err_{f_e}\right)$ and $h\left(err_{f_t}\right)$ for each region and **(h)** the corresponding mean values for the period 2015–2017. In panels **(a)**–**(g)**, the upper-right corner displays a number indicating the correlation coefficient between $h\left(err_{f_e}\right)$ and $h\left(err_{f_t}\right)$. The shaded areas and error bars represent 95 % confidence intervals derived from 1000 bootstrap samples of datasets.

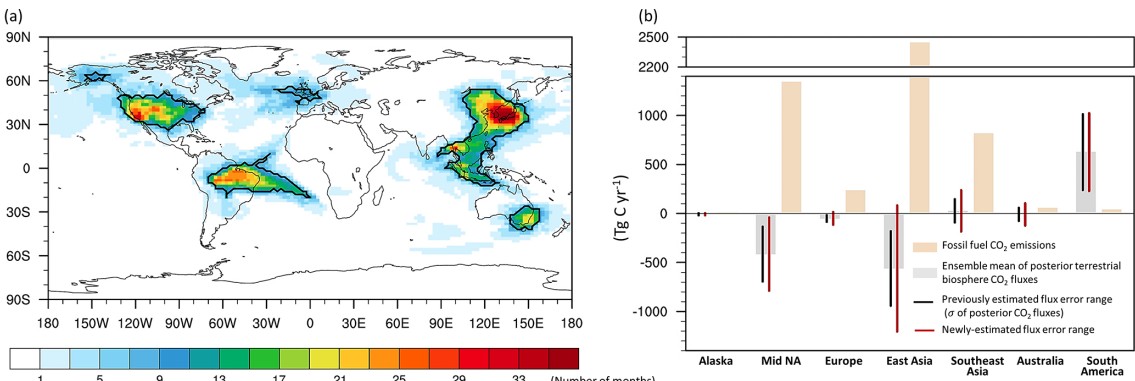

**Figure 6. (a)** Number of months selected as the effective areas for airborne measurements. The outlined areas represent selected areas for 8 months or more. **(b)** Annual total terrestrial-biosphere CO$_2$ fluxes obtained from the ensemble mean of 10 OCO-2 MIP models and annual total fossil fuel CO$_2$ emissions estimated from ODIAC data for each outlined area, averaged over the period 2015–2017. The black error bars each represent $\pm 1$ standard deviation of the posterior net land fluxes, identical to the standard deviation of the posterior terrestrial-biosphere fluxes. The red error bars indicate newly estimated ranges of errors in the posterior net land fluxes from this study.

when utilizing airborne CO$_2$ measurements (and potentially other CO$_2$ observations) to analyze the detailed characteristics of ensemble posterior flux estimates at a regional (or latitudinal) level, it is crucial to account for the contributions of representation and transport errors.

Our analysis reveals that true errors in the ensemble mean of the posterior net CO$_2$ flux estimates are significantly greater than the ensemble spread of the flux estimates in five out of the seven regions where fossil fuel emissions are higher than terrestrial-biosphere fluxes. A possible explanation for this result is the presence of errors in the prescribed fossil fuel emissions common to all OCO-2 MIP models. The OCO-2 MIP models treated fossil fuel emissions as perfectly known values and adjusted terrestrial-biosphere and ocean CO$_2$ fluxes to minimize the difference between the simulated and observed CO$_2$ concentrations. Thus, if there are errors in the prescribed fossil fuel emission estimates, these errors propagate into the posterior natural flux estimates. The assumption used in the OCO-2 MIP models is, in fact, one that is often applied in conventional global atmospheric inverse models as errors in fossil fuel emission estimates are considered to be lower than those in natural flux estimates at national scales (4 %–20 %; Andres et al., 2014). However, the emission errors become substantial when considering spatial distributions at model grid scales and temporal variability over a year (Zhang et al., 2016; Gurney et al., 2021). Oda et al. (2023) showed the significant impacts of differences in fossil fuel emission estimates on posterior terrestrial-biosphere flux estimates near the source regions. The OCO-2 MIP models used identical fossil fuel emission estimates, and thus their posterior net flux estimates share common biases, induced by errors in the fossil fuel emission estimates. Because these systematic biases are not captured by the ensemble spread of the flux estimates, the true flux errors exceed the errors computed from the ensemble spread

in the main source regions. In addition, the regional and seasonal sampling biases of CO$_2$ measurements and satellite retrieval errors may contribute to these systematic biases (Kulawik et al., 2019). Using eight prior flux datasets may also not adequately represent the errors in the terrestrial-biosphere fluxes, which exhibit significant variations across the estimates (Feng et al., 2019). Therefore, further studies are needed to uncover the causes of underestimation in true flux errors and thus to understand the uncertainty sources overlooked in current ensemble inverse-modeling estimates.

The reliability of our observation-based regional flux error estimates depends on the availability of airborne measurement data. Although our approach is generally effective for estimating regional means of monthly $h\left(\mathrm{err}_{\mathrm{f_t}}\right)$ values, it is not applicable in 15 %$^{TS1}$ of our cases (shown in Fig. 5), where measurements were mostly taken in local areas covering one to six $1° \times 1°$ grid cells within each region. This limitation may be attributed to the application of a common method for calculating observation errors across all data points, which might not adequately identify specific outliers. Caution is required when applying our approach to monthly-scale analyses, especially when using observations made locally. Extending the calculation period to several months or longer (e.g., Fig. 5h) is a suitable strategy for mitigating the impact of outliers and obtaining more robust results. In fact, the ratios between the 3-year mean values of $h\left(\mathrm{err}_{\mathrm{f_e}}\right)$ and $h\left(\mathrm{err}_{\mathrm{f_t}}\right)$, which serve as key metrics for quantifying regional flux errors (Fig. 5h), exhibit smaller uncertainties over midlatitude North America and East Asia, where consistent airborne data covering broad areas are available, than over Europe and South America, where airborne observations are sparse and data coverage is intermittent. In addition, it is noteworthy that the $h\left(\mathrm{err}_{\mathrm{f_e}}\right)$-to-$h\left(\mathrm{err}_{\mathrm{f_t}}\right)$ ratios derived from continuous observations enable the computation of unbiased true errors in the ensemble mean of annual posterior net

fluxes, averaged over the analysis period, compared to those derived from limited observation periods (e.g., for Alaska). These results highlight the importance of frequent airborne measurements with extensive spatial coverage for reliably quantifying errors in regional net flux estimates derived from inverse models.

The performance of inverse models in simulating atmospheric $CO_2$ may vary by season. However, airborne measurements were not uniformly conducted across all seasons in most analyzed regions. Among the seven regions analyzed, East Asia was monitored by the CONTRAIL program, which continuously conducted $CO_2$ measurements over 3 years, with routes repeated throughout all seasons. This resulted in the area most sensitive to the measurements exhibiting similar spatial patterns during the vegetation growing season (from May to October) and the non-growing season, covering the northeastern part of China, Korea, and Japan (Fig. S7). The airborne measurements taken in East Asia offer a unique opportunity to explore seasonal variations in regional error statistics. For the period 2015–2017, the regional averages of both the RMSE and $ERR_{TOT}$ are, on average, 12 % and 11 % higher, respectively, during the non-growing season than during the growing season (Fig. S8). In contrast, the regional averages of $h\left(err_{f_e}\right)$ and $h\left(err_{f_t}\right)$ are higher during the growing season (0.90 [0.84, 0.97] and 1.37 [1.13, 1.62] ppm, respectively) than during the non-growing season (0.66 [0.62, 0.70] and 1.30 [1.06, 1.54] ppm, respectively) because $CO_2$ errors tend to increase proportionally with the magnitude of the flux values. Consequently, the ratio of $h\left(err_{f_e}\right)$ to $h\left(err_{f_t}\right)$ is slightly lower during the non-growing season, with a value of 0.51 [0.39, 0.64], than during the growing season, with a value of 0.66 [0.50, 0.83], indicating a greater underestimation of true flux errors when the terrestrial-biosphere $CO_2$ sinks are relatively smaller. This result aligns with our finding that true net land flux errors are significantly underestimated when fossil fuel emissions are larger in magnitude than terrestrial-biosphere fluxes. Furthermore, the consistent ratio of $h\left(err_{f_e}\right)$ to $h\left(err_{f_t}\right)$ below 1, without significant seasonal variations in East Asia, suggests that our conclusions, drawn from the analysis of seven regions, may not be seasonally dependent.

To capture the signals from regional surface $CO_2$ fluxes, we use atmospheric $CO_2$ data observed and simulated within the 1–5 km a.g.l. altitude range. The choice of this altitude range may have influenced the regional error statistics as the performance of the inverse models could vary with altitude. To gauge this sensitivity, we compare error statistics derived from atmospheric $CO_2$ data across two altitude ranges: 1–3 and 1–5 km a.g.l. Among the seven analyzed regions, Australia and South America are excluded from this additional analysis because the airborne observations for these two regions cover fewer than 100 grid cells with respect to the analysis period, and narrowing the altitude range resulted in the loss of over 30 % of the grid cells. The areas sensitive to airborne $CO_2$ measurements within the two altitude ranges exhibit nearly identical spatial patterns across Alaska, midlatitude North America, Europe, East Asia, and Southeast Asia, indicating that observations corresponding to lower altitudes are more sensitive to surface $CO_2$ fluxes (Fig. S9). Because of the higher sensitivity, error statistics for all regions have larger values when calculated using data from the 1–3 km a.g.l. altitude range than when determined using data from the 1–5 km a.g.l. altitude range (Fig. S10). For example, in midlatitude North America, the regional averages of the RMSE, $ERR_{TOT}$, $h\left(err_{f_e}\right)$, and $h\left(err_{f_t}\right)$ are 1.42 [1.36, 1.49], 1.34 [1.30, 1.39], 0.72 [0.69, 0.76], and 0.86 [0.72, 1.01] ppm, respectively, when calculated using data from the 1–3 km a.g.l. altitude range. In comparison, when computed using data from the 1–5 km a.g.l. altitude range, these values are 1.21 [1.15, 1.26], 1.09 [1.06, 1.13], 0.57 [0.55, 0.60], and 0.77 [0.66, 0.88] ppm, respectively. However, the ratio between the 3-year mean values of $h\left(err_{f_e}\right)$ and $h\left(err_{f_t}\right)$ does not show significant differences based on the altitude ranges, with differences ranging from 0.02 to 0.11. Again, these results suggest that our observation-based regional flux error estimates are not sensitive to the choice of altitude range over longer time periods.

Our study computes the true flux errors for the ensemble mean estimates by comparing $RMSE^2$ and $ERR_{TOT}^2$. However, discrepancies between the true and estimated values of observation, representation, and transport errors, as well as covariances between flux errors and transport errors, may contribute to variations in $RMSE^2$ and $ERR_{TOT}^2$. Due to a lack of information across all datasets, we set the observation errors under ideal conditions (i.e., 0.1 ppm). In reality, inadequate quality control could have resulted in significant systematic biases for specific regions and time periods (Masarie et al., 2011; Baier et al., 2020), impacting our results, especially those concerning South America. For instance, if the average measurement error had been 0.5 ppm instead of the assumed value of 0.1 ppm during the analysis period, the calculated true flux error would have decreased from 398 to 334 $Tg C yr^{-1}$ for South America and from 374 to 260 $Tg C yr^{-1}$ for midlatitude North America.

Representation errors and $h\left(err_{f_e}\right)$ are derived using the GEOS-5 and GEOS-Chem models, but these values depend on the transport model and meteorological fields used. Employing our approach across all participating MIP models to compute these two error terms and subsequently averaging them would lead to more realistic flux error quantification in future studies. Employing all transport models also would facilitate the calculation of variances of flux errors and their covariance with transport errors included in $ERR_{MIP}$, as shown in Appendix A, and subsequently enable the determination of the total true flux errors, including both diagonal and off-diagonal terms. In addition, previous studies show that 8–10 different ensemble members are required for robust transport error estimates (Feng et al., 2019; Lauvaux et al., 2019). However, out of the 10 ensemble members in the OCO-2 MIP, 3 employed TM5 and 5 utilized GEOS-Chem

(Table S1). This ensemble size may not be sufficient for fully capturing the range of true transport errors. We further investigate how our main results would be affected if the estimated transport errors deviated by 20 % or 40 % from the actual errors, based on the difference between $RMSE^2$ and $ERR_{TOT}^2$. The ratio of the regional mean of $h\left(err_{f_e}\right)$ to that of $h\left(err_{f_t}\right)$ increases, on average, by only up to 0.04 and 0.09, respectively, in the seven regions throughout the analysis period (Fig. S11). In both cases, the estimated flux errors for midlatitude North America, Europe, East Asia, and Southeast Asia still show significant underestimations at a 95 % confidence level, whereas this is not observed for Alaska and South America. In Australia, characterized by a wide uncertainty range, significant underestimation is also observed in the "20 % deviation" cases, supporting the robustness of our findings. In future versions of the OCO-2 MIP, the participation of inverse-modeling groups using other transport models or meteorological forcing data might contribute to the estimation of transport errors closer to actual values.

This study uses monthly posterior flux estimates for the calculation of monthly $h\left(err_{f_e}\right)$ values. However, posterior flux estimates from each OCO-2 MIP model have different sub-monthly patterns, which could modify the sub-monthly variations in posterior atmospheric $CO_2$ concentrations and, in turn, affect their ensemble spread. To examine their potential impact on the results, we conduct an analysis with different publicly available hourly (or 3-hourly) terrestrial-biosphere fluxes (Chevallier et al., 2019; Jacobson et al., 2020; Ott, 2020; Haynes et al., 2021; Liu and Bowman, 2024), which are from seven OCO-2 MIP prior flux models (Ames, Baker, CAMS, CMS-Flux, CT, OU, and WOMBAT; Table S1). By incorporating the monthly balanced hourly flux estimates into the monthly posterior fluxes, we generate hourly posterior terrestrial-biosphere flux estimates for these seven models. Since the assimilation window for each OCO-2 MIP model ranges from 1 week to 1 month, the weekly variations in the posterior fluxes may differ from those in the prior fluxes. Nonetheless, with only the monthly posterior flux estimates being publicly available, this approach offers valuable insights into how different sub-monthly patterns of posterior fluxes might affect our main results. Our analysis shows that the regional averages of $h\left(err_{f_e}\right)$ derived from the monthly posterior flux estimates from the 7 models are, on average, within ±10 % of the values originally obtained using flux estimates from all 10 models for the period 2015–2017, except with respect to Europe (13 % lower) (Fig. S12a). When accounting for different sub-monthly patterns of posterior fluxes across models, the regional averages of $h\left(err_{f_e}\right)$ increase by 10 %–22 % (0.06–0.14 ppm) across six regions, with a 45 % (0.23 ppm) increase for Europe. These results suggest that our earlier calculation, assuming identical sub-monthly flux variations, underestimates $h\left(err_{f_e}\right)$. We further investigated whether our main finding remains robust even when adjusting the original values of $h\left(err_{f_e}\right)$ using the potential-underestimation

rate. After making this correction, we found that the ratios between the regional average values of $h\left(err_{f_e}\right)$ and $h\left(err_{f_t}\right)$ increased the most in Europe (by 0.14) and by only up to 0.07 in the other six regions as $h\left(err_{f_t}\right)$ also increases with $h\left(err_{f_e}\right)$ according to Eq. (8) (Fig. S12b). Moreover, $h\left(err_{f_t}\right)$ still exhibits significant underestimation ($p < 0.05$) in midlatitude North America, Europe, East Asia, Southeast Asia, and Australia. This indicates that our main results are robust to the inclusion or exclusion of sub-monthly flux patterns in the calculation of $h\left(err_{f_e}\right)$.

In summary, our study provides an observation-based method for quantifying errors in the ensemble mean of regional net $CO_2$ flux estimates, which can be widely applied in inverse model intercomparison projects, such as the OCO-2 MIP. The evaluation results of the OCO-2 MIP ensemble members reveal that the true errors in ensemble posterior fluxes are larger than the ensemble spread in regions with high anthropogenic $CO_2$ emissions. This result provides observation-based evidence supporting previous studies (Wang et al., 2020; Oda et al., 2023) that emphasized the impact of fossil fuel emission errors on global atmospheric $CO_2$ inversions. This finding offers important insights into understanding the sources of errors in current inverse modeling and highlights the need for improving fossil fuel emission estimates and developing inversion methods that optimize both fossil fuel emissions and natural fluxes. Airborne observations provide a broader footprint compared to ground-based observations. Leveraging this advantage, our study evaluates 19 % of the total global land cover (excluding Antarctica and Greenland), but data scarcity limits the evaluation of the remaining 81 %. In addition to ongoing airborne measurement programs – including the CONTRAIL and IAGOS-CARIBIC programs, as well as various airborne programs hosted by INPE, NASA, and NOAA – airborne observations have been conducted in unexplored regions, including Siberia (e.g., Narbaud et al., 2023), Africa (e.g., Barker et al., 2020), and northern Europe (e.g., Barker et al., 2021). Sustained efforts to maintain and expand airborne observations, along with a collaborative data-sharing and management system (e.g., ObsPack), will contribute to accurately estimating and reducing uncertainties in regional surface $CO_2$ fluxes.

## Appendix A

Following Eq. (1) in the main text,

$$RMSE^2 = \frac{1}{N} \sum_{i=1}^{N} \left[ y_{o,i} - \overline{h\left(\hat{x}_i\right)} \right] \left[ y_{o,i} - \overline{h\left(\hat{x}_i\right)} \right]^T,$$

$$\text{where } \overline{h(\hat{x}_i)} = \frac{1}{M} \sum_{j=1}^{M} h_j \left( \hat{x}_{j,i} \right). \quad (A1)$$

Here, $\overline{h(\hat{x}_i)}$ denotes the ensemble mean of the posterior $CO_2$ concentrations from the OCO-2 MIP models corresponding to the $i$th airborne observation ($y_{o,i}$) within each $1° × 1°$ grid

cell for each month. $N$ is the total number of data points from airborne measurements sampled in each grid cell monthly. $M$ is the ensemble size (i.e., 10 members).

Equation (A1) can be rewritten as TS2

$$\text{RMSE}^2 = \frac{1}{N}\sum_{i=1}^{N}\left[\left(y_{\text{o},i} - h_{\text{t}}\left(\hat{x}_{\text{t},i}\right)\right) - \left(\overline{h\left(\hat{x}_i\right)} - h_{\text{t}}\left(\hat{x}_{\text{t},i}\right)\right)\right]$$
$$\left[\left(y_{\text{o},i} - h_{\text{t}}\left(\hat{x}_{\text{t},i}\right)\right) - \left(\overline{h\left(\hat{x}_i\right)} - h_{\text{t}}\left(\hat{x}_{\text{t},i}\right)\right)\right]^T \quad \text{(A2)}$$

$$= \frac{1}{N}\sum_{i=1}^{N}\left[y_{\text{o},i} - h_{\text{t}}\left(\hat{x}_{\text{t},i}\right)\right]\left[y_{\text{o},i} - h_{\text{t}}\left(\hat{x}_{\text{t},i}\right)\right]^T$$
$$- 2\left(y_{\text{o},i} - h_{\text{t}}\left(\hat{x}_{\text{t},i}\right)\right)\left(\overline{h\left(\hat{x}_i\right)} - h_{\text{t}}\left(\hat{x}_{\text{t},i}\right)\right)$$
$$+ \left[\overline{h\left(\hat{x}_i\right)} - h_{\text{t}}\left(\hat{x}_{\text{t},i}\right)\right]\left[\overline{h\left(\hat{x}_i\right)} - h_{\text{t}}\left(\hat{x}_{\text{t},i}\right)\right]^T,$$
$$\text{(A3)}$$

where $h_{\text{t}}(\hat{x}_{\text{t}})$ denotes the estimated $CO_2$ concentration obtained from an error-free atmospheric transport model ($h_{\text{t}}$) and a set of true $CO_2$ fluxes ($\hat{x}_{\text{t}}$). The three terms on the right-hand side of Eq. (A3) represent (i) the variances of observation and representation errors, (ii) the covariances between errors in observation and representation and errors in flux and transport, and (iii) the variances of flux and transport errors in the ensemble estimates. Assuming the independence of observation and representation errors from transport and flux errors, Eq. (A3) can be simplified as

$$\text{RMSE}^2 = \frac{1}{N}\sum_{i=1}^{N}\left[y_{\text{o},i} - h_{\text{t}}\left(\hat{x}_{\text{t},i}\right)\right]\left[y_{\text{o},i} - h_{\text{t}}\left(\hat{x}_{\text{t},i}\right)\right]^T$$
$$+ \left[\overline{h\left(\hat{x}_i\right)} - h_{\text{t}}\left(\hat{x}_{\text{t},i}\right)\right]\left[\overline{h\left(\hat{x}_i\right)} - h_{\text{t}}\left(\hat{x}_{\text{t},i}\right)\right]^T. \quad \text{(A4)}$$

Further, the second term on the right-hand side of Eq. (A4) can be rewritten by separating the flux error and transport error terms as follows:

$$\frac{1}{N}\sum_{i=1}^{N}\left[\overline{h\left(\hat{x}_i\right)} - h_{\text{t}}\left(\hat{x}_{\text{t},i}\right)\right]\left[\overline{h\left(\hat{x}_i\right)} - h_{\text{t}}\left(\hat{x}_{\text{t},i}\right)\right]^T$$
$$= \frac{1}{N}\sum_{i=1}^{N}\left[\left(\overline{h\left(\hat{x}_i\right)} - \overline{h\left(\hat{x}_{\text{t},i}\right)}\right) - \left(h_{\text{t}}\left(\hat{x}_{\text{t},i}\right) - \overline{h\left(\hat{x}_{\text{t},i}\right)}\right)\right]$$
$$\left[\left(\overline{h\left(\hat{x}_i\right)} - \overline{h\left(\hat{x}_{\text{t},i}\right)}\right) - \left(h_{\text{t}}\left(\hat{x}_{\text{t},i}\right) - \overline{h\left(\hat{x}_{\text{t},i}\right)}\right)\right]^T \quad \text{(A5)}$$
$$= \frac{1}{N}\sum_{i=1}^{N}\left[\overline{h\left(\hat{x}_i\right)} - \overline{h\left(\hat{x}_{\text{t},i}\right)}\right]\left[\overline{h\left(\hat{x}_i\right)} - \overline{h\left(\hat{x}_{\text{t},i}\right)}\right]^T$$
$$- 2\left(\overline{h\left(\hat{x}_i\right)} - \overline{h\left(\hat{x}_{\text{t},i}\right)}\right)\left(h_{\text{t}}\left(\hat{x}_{\text{t},i}\right) - \overline{h\left(\hat{x}_{\text{t},i}\right)}\right)$$
$$+ \left[h_{\text{t}}\left(\hat{x}_{\text{t},i}\right) - \overline{h\left(\hat{x}_{\text{t},i}\right)}\right]\left[h_{\text{t}}\left(\hat{x}_{\text{t},i}\right) - \overline{h\left(\hat{x}_{\text{t},i}\right)}\right]^T. \quad \text{(A6)}$$

The three terms on the right-hand side of Eq. (A6) represent (i) the variances of flux errors in concentration space, (ii) the covariances between flux errors and transport errors, and (iii) the variances of transport errors.

In the OCO-2 MIP, by approximating the ensemble spread of the posterior fluxes as true errors in the mean fluxes, it is assumed that the values of the first and second terms on the right-hand side of Eq. (A4) can be written as the sum of the

set of observation errors ($\text{ERR}_{\text{OBS}}^2$), the set of representation errors ($\text{ERR}_{\text{REP}}^2$), and the ensemble spread of posterior $CO_2$ concentrations across the OCO-2 MIP models ($\text{ERR}_{\text{MIP}}^2$):

$$\text{RMSE}^2 \approx \text{ERR}_{\text{TOT}}^2 = \text{ERR}_{\text{OBS}}^2 + \text{ERR}_{\text{REP}}^2 + \text{ERR}_{\text{MIP}}^2. \quad \text{(A7)}$$

We assume that the observation errors are independent of the representation errors.

$\text{ERR}_{\text{MIP}}^2$ can be also rewritten by separating the flux error and transport error terms as follows:

$$\text{ERR}_{\text{MIP}}^2 = \frac{1}{N}\sum_{i=1}^{N}\frac{1}{M}\sum_{j=1}^{M}$$
$$\left[\overline{h\left(\hat{x}_i\right)} - h_j\left(\hat{x}_{j,i}\right)\right]\left[\overline{h\left(\hat{x}_i\right)} - h_j\left(\hat{x}_{j,i}\right)\right]^T \quad \text{(A8)}$$
$$= \frac{1}{N}\sum_{i=1}^{N}\frac{1}{M}\sum_{j=1}^{M}\frac{1}{M}\sum_{k=1}^{M}$$
$$\left[\left(h_k\left(\hat{x}_{k,i}\right) - h_k\left(\hat{x}_{j,i}\right)\right) - \left(h_j\left(\hat{x}_{j,i}\right) - h_k\left(\hat{x}_{j,i}\right)\right)\right]$$
$$\left[\left(h_k\left(\hat{x}_{k,i}\right) - h_k\left(\hat{x}_{j,i}\right)\right) - \left(h_j\left(\hat{x}_{j,i}\right) - h_k\left(\hat{x}_{j,i}\right)\right)\right]^T \quad \text{(A9)}$$
$$= \frac{1}{N}\sum_{i=1}^{N}\frac{1}{M}\sum_{k=1}^{M}\frac{1}{M}\sum_{j=1}^{M}$$
$$\left[h_k\left(\hat{x}_{k,i}\right) - h_k\left(\hat{x}_{j,i}\right)\right]\left[h_k\left(\hat{x}_{k,i}\right) - h_k\left(\hat{x}_{j,i}\right)\right]^T$$
$$- 2\left(h_k\left(\hat{x}_{k,i}\right) - h_k\left(\hat{x}_{j,i}\right)\right)\left(h_j\left(\hat{x}_{j,i}\right) - h_k\left(\hat{x}_{j,i}\right)\right)$$
$$+ \left[h_j\left(\hat{x}_{j,i}\right) - h_k\left(\hat{x}_{j,i}\right)\right]\left[h_j\left(\hat{x}_{j,i}\right) - h_k\left(\hat{x}_{j,i}\right)\right]^T.$$
$$\text{(A10)}$$

As in Eq. (A6), the three terms on the right-hand side of Eq. (A10) correspond to the (i) approximated variances of flux errors, (ii) the approximated covariances between flux errors and transport errors, and (iii) the approximated variances of transport errors. For the calculation of the first term, utilizing all participating transport models from the OCO-2 MIP would be ideal; however, in this study, we approximate it using the GEOS-Chem model.

**Code and data availability.** The inverse-modeling results and airborne $CO_2$ measurement data from the OCO-2 v10 MIP are available on the official website of the NOAA/ESRL Global Monitoring Laboratory at https://www.gml.noaa.gov/ccgg/OCO2_v10mip/download.php (NOAA/ESRL Global Monitoring Laboratory, 2022). The high-resolution global GEOS-Chem simulation results used to calculate the representation errors can be obtained upon request from Brad Weir (brad.weir@nasa.gov) or Lesley Ott (lesley.e.ott@nasa.gov). The forward and adjoint sensitivity simulations for this work were conducted using the publicly available GEOS-Chem adjoint model. This model can be downloaded from http://wiki.seas.harvard.edu/geos-chem/index.php/GEOS-Chem_Adjoint (Henze et al., 2007). ODIAC fossil fuel $CO_2$ emission data are available at https://doi.org/10.17595/20170411.001 (Oda and Maksyutov, 2015). Hourly (or 3-hourly) terrestrial-biosphere carbon flux datasets from CASA-GFED3, CASA-GFED4.1s, SiB4, CAR-DAMOM, and ORCHIDEE – all OCO-2 v10 MIP prior flux models – are available at https://doi.org/10.5067/VQPRALE26L20 (Ott, 2020), https://doi.org/10.25925/20201008 (Jacobson et al.,

2020), https://doi.org/10.3334/ORNLDAAC/1847 (Haynes et al., 2021), https://doi.org/10.5067/1XO0PZEZOR1H (Liu and Bowman, 2024), and on the official website of the CAMS (https://ads.atmosphere.copernicus.eu/datasets/cams-global-greenhouse-gas-inversion?tab=download, Copernicus Atmosphere Monitoring Service, 2020).

**Supplement.** The supplement related to this article is available online at: https://doi.org/10.5194/acp-25-1-2025-supplement.

**Author contributions.** JY and JL designed this study, and JY performed the analysis. JL, BrB, BW, KM, and BiB reviewed the paper and provided input. BW and LEO provided the high-resolution global GEOS-Chem simulation results. KM, BiB, LVG, and SCB provided the airborne CO$_2$ observations. JY led the writing, with input from all coauthors.

**Competing interests.** The contact author has declared that none of the authors has any competing interests.

**Disclaimer.** Publisher's note: Copernicus Publications remains neutral with regard to jurisdictional claims made in the text, published maps, institutional affiliations, or any other geographical representation in this paper. While Copernicus Publications makes every effort to include appropriate place names, the final responsibility lies with the authors.

**Acknowledgements.** We are grateful to NOAA, NASA LaRC, NASA GSFC, INPE, NIES, MRI, the European research infrastructure IAGOS-CARIBIC, the DOE's LBNL, and Harvard University for providing the airborne CO$_2$ observation data. We thank Colm Sweeney, Kenneth Davis, Joshua P. DiGangi, Melissa Y. Martin, John B. Miller, Emanuel Gloor, Wouter Peters, Toshinobu Machida, Hidekazu Matsueda, Yousuke Sawa, Yosuke Niwa, Ed Dlugokencky, Stephan R. Kawa, James B. Abshire, Haris Riris, Florian Obersteiner, Harald Boenisch, Torsten Gehrlein, Andreas Zahn, Christoph Gerbig, Tanja Schuck, Gao Chen, Michael Shook, Giordane A. Martins, Rodrigo A. de Souza, Britton Stephens, Eric Kort, Thomas Ryerson, Jeff Peischl, Ken Aikin, Steve Wofsy, Bruce Daube, and Roisin Commane for contributing airborne CO$_2$ data. We also appreciate all the providers of the datasets used in the GEOS-Chem model simulations. We especially thank all research groups involved in the OCO-2 MIP for their contributions to providing v10 OCO-2 and in situ CO$_2$ datasets, as well as inverse-modeling outputs.

**Financial support.** This research was carried out at the Jet Propulsion Laboratory, California Institute of Technology, under a contract with the National Aeronautics and Space Administration (80NM0018D0004) and has been supported by the National Aeronautics and Space Administration (grant no. 20-OCOST20-0012). The South American airborne CO$_2$ measurements were supported by the European Research Council through the ASICA project (grant no. 649087). Observations collected at the SGP site were partly supported by the Office of Biological and Environmental Research of the US Department of Energy under contract no. DE-AC02-05CH11231 as part of the Atmospheric Radiation Measurement (ARM) program, the ARM Aerial Facility, and the Environmental System Science (ESS) program.

**Review statement.** This paper was edited by Christoph Gerbig and reviewed by two anonymous referees.

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

## Remarks from the typesetter