# Peer review of "Quantification of regional net CO2 flux errors in the v10 OCO-2 MIP ensemble using airborne measurements"

_EGUsphere, 2023_

## Author Comment (AC1)

[Reply] We appreciate your constructive comments on this manuscript. We revised the manuscript to fully address your comments and suggestions. Detailed point-by-point responses to your comments and related revisions are presented below. The original comments are in black, and our responses are in blue color.

1. The OCO-2 v10 MIP sampled a much wider set of aircraft data than those used in this study. In particular NOAA operates a light aircraft program that produces regular profiles of CO2 measurements over North America and Raratonga. These data should be well suited to the analysis conducted here due to the regular sampling frequency, nearly continuous coverage, and altitudes sampled. For some reason, of these timeseries stations, only the data from Dahlen, North Dakota (DND) and Marcellus, Pennsylvania (MRC) were included in Table 1 of the manuscript. In addition to these two sites, there are evaluation data in the OCO-2 MIP samples from timeseries over:
Briggsdale, Colorado - (CAR)
Offshore Cape May, New Jersey - (CMA)
Carbon in Arctic Reservoirs Vulnerability Experiment (CARVE) - (CRV)
Estevan Point,  British Columbia - (ESP)
East Trout Lake, Saskatchewan - (ETL)
Homer, Illinois - (HIL)
INFLUX (Indianapolis Flux Experiment) - (INX)
Park Falls, Wisconsin - (LEF)
Offshore Portsmouth, New Hampshire (Isles of Shoals) - (NHA)
Poker Flat, Alaska - (PFA)
Rarotonga - (RTA)
Offshore Charleston, South Carolina - (SCA)
Southern Great Plains, Oklahoma - (SGP)
Offshore Corpus Christi, Texas - (TGC)
Trinidad Head, California - (THD)
West Branch, Iowa - (WBI)

[Reply] We appreciate your suggestions. In this study, only measurement data not assimilated in the LNLGIS experiment were utilized for analysis. Additionally, we required a minimum of 10 observations per month at each 1°x1° grid point to calculate error statistics. Consequently, many of the airborne measurement data in OCO-2 MIP datasets did not meet our analysis criteria. We re-evaluated the availability of the dataset and the list of data used (Table 1 of the revised manuscript; shown in Table R1) to include all airborne measurements that meet our standards at least once. Through this process, CAR, PFA, and SGP data were newly incorporated into the analysis, while data that did not meet the criteria, AOA and MRC, were excluded from the data list. The addition of these three data sets did not result in any noticeable changes in our results.

**Table R1** Data description for each airborne measurement campaign.

| Site code | Site name | Measurement campaign name | Measurement type | Data provider | ObsPack (*original*) dataset identifier | Reference |
|---|---|---|---|---|---|---|
| **ACG** | Alaska Coast Guard | NOAA/GML Aircraft Program | In situ | National Oceanic and Atmospheric Administration (NOAA) Global Monitoring Laboratory (GML) | http://doi.org/10.259 25/20201204[a] | |
| **ACT** | Atmospheric Carbon and Transport – America (ACT-America) | ACT-America | In situ and flask | NASA Langley Research Center (NASA-LaRC), NOAA/GML | http://doi.org/10.259 25/20201204[a] *https://doi.org/10.33 34/ORNLDAAC/1593* | Baier et al. (2020) Wei et al. (2021) |

| | | | | | | |
|---|---|---|---|---|---|---|
| **AirCore NOAA** | NOAA AirCore | NOAA AirCore Program | Balloon | NOAA/GML | No Obspack DOI[b] *https://doi.org/10.15138/6AV0-MY81* | Karion et al. (2010) |
| **ALF** | Alta Floresta | | Flask | National Institute for Space Research (INPE) | http://dx.doi.org/10.25925/20181030[c] https://doi.org/10.1594/PANGAEA.926834 | Gatti et al. (2023) |
| **CAR** | Briggsdale, Colorado | | Flask | NOAA/GML | http://doi.org/10.25925/20210517[d] | Sweeney et al. (2015) |
| **CON** | Comprehensive Observation Network for TRace gases by AIrLiner (CONTRAIL) | | In situ | National Institute for Environmental Studies (NIES), Meteorological Research Institute (MRI) | http://doi.org/10.25925/20201204[a] *https://doi.org/10.17595/20180208.001* | Machida et al. (2008) |
| **CRV** | Carbon in Arctic Reservoirs Vulnerability Experiment (CARVE) | Arctic-Boreal Vulnerability Experiment (ABoVE) | In situ | NOAA/GML | http://doi.org/10.25925/20201204[a] *https://doi.org/10.3334/ORNLDAAC/1582* | |
| **GSFC** | NASA Goddard Space Flight Center Aircraft Campaign | | In situ | NASA Goddard Space Flight Center (NASA-GSFC) | http://doi.org/10.25925/20201204[a] | Kawa et al. (2018) |
| **IAGOS** | In-service Aircraft for a Global Observing System | Civil Aircraft for the Regular Investigation of the atmosphere Based on an Instrument Container (IAGOS-CARIBIC) | In situ | Karlsruhe Institute of Technology (IMK-ASF), Institute for Atmospheric and Environmental Sciences (IAU), Max Planck Institute for Biogeochemistry (MPI-BGC) | http://doi.org/10.25925/20201204[a] | Filges et al. (2015) |
| **LARC** | LARC - NASA Langley Research Center Aircraft Campaign | Korea-United States Air Quality Study | In situ | NASA-LaRC | http://doi.org/10.25925/20201204[a] | |
| **MAN** | Manaus | | In situ | NOAA/GML | https://doi.org/10.25925/20210519[e] | |
| **ORC** | O2/N2 Ratio and CO2 Airborne Southern Ocean Study (ORCAS) | | In situ | National Center for Atmospheric Research (NCAR) | http://doi.org/10.25925/20201204[a] *https://doi.org/10.5065/D6SB445X* | Stephens et al. (2018) |
| **PAN** | Pantanal, Mato Grosso do Sul | | Flask | INPE | http://dx.doi.org/10.25925/20181030[c] | |
| **PFA** | Poker Flat, Alaska | | Flask | NOAA/GML | http://doi.org/10.25925/20210517[d] | Sweeney et al. (2015) |
| **RBA-B** | Rio Branco | | Flask | INPE | http://dx.doi.org/10.25925/20181030[c] *https://doi.org/10.1594/PANGAEA.926834* | Gatti et al. (2023) |
| **SAN** | Santarem | | Flask | INPE | http://dx.doi.org/10.25925/20181030[c] *https://doi.org/10.1594/PANGAEA.926834* | Gatti et al. (2023) |
| **SGP** | Southern Great Plains, Oklahoma | | Flask | The US Department of Energy (DOE)/Lawrence Berkeley National Laboratory (LBNL) | http://doi.org/10.25925/20210517[d] | Biraud et al. (2013) |
| **SONGNEX2015** | Shale Oil and Natural Gas Nexus 2015 (air campaign) | | In situ | NOAA Chemical Sciences Laboratory (CSL) | http://doi.org/10.25925/20201204[a] | |
| **TEF** | Tefe | | Flask | INPE | http://dx.doi.org/10.25925/20181030[c] *https://doi.org/10.1594/PANGAEA.926834* | Gatti et al. (2023) |
| **TOM** | Atmospheric Tomography Mission (ATom) | | In situ | NOAA/GML, Harvard University | http://doi.org/10.25925/20201204[a] *https://doi.org/10.3334/ORNLDAAC/1581* | |

ᵃ: obspack_co2_1_GLOBALVIEWplus_v6.1_2021-03-01 (Schuldt et al., 2021b)
ᵇ: obspack_co2_1_AirCore_v4.0_2020-12-28
ᶜ: obspack_co2_1_INPE_RESTRICTED_v2.0_2018-11-13 (NOAA Carbon Cycle Group ObsPack Team, 2018)
ᵈ: obspack_co2_1_NRT_v6.1.1_2021-05-17 (Schuldt et al., 2021a)
ᵉ: obspack_multi-species_1_manaus_profiles_v1.0_2021-05-20 (Miller et al., 2021)

2. This reviewer's experience with simulation of aircraft measurements is that model residuals are strongly affected by altitude and by season. The analysis here does not discriminate by either of these factors, except to choose an altitude range apparently chosen to minimize the effect of residuals closer to the surface. Should the model residuals have significant variability by these factors, the evaluation criteria would be affected and possibly dominated by those factors, which would confound the statistical conclusions of this work. I suggest that a factor analysis, possibly an analysis of variance, is needed to determine whether model residuals are driven by these factors.

[Reply] Based on the reviewer's suggestions, we conducted additional analyses to explore how our results vary seasonally and with changes in the chosen altitude range.

First, to isolate the seasonal impacts on the regional error statistics (e.g., RMSE, ERR$_{TOT}$, $h\left(err_{f_e}\right)$, and $h\left(err_{f_t}\right)$), it is essential that other factors influencing the error quantities, such as the number of observation points and observation coverage within each region, remain consistent across seasons. Among the seven regions analyzed, in East Asia, the CONTRAIL program has continuously conducted measurements over three years with routes repeated throughout all seasons. The total numbers of observed grid-points per month during the vegetation growing season (from May to October) and non-growing season for 2015−2017 are comparable, amounting to 404 and 428, respectively. Furthermore, the area most sensitive to airborne measurements exhibit similar spatial patterns in both seasons, encompassing the northeast part of China, the Korean Peninsula, and Japan (Figure R1). By focusing on this region, we examined how the error quantities vary by seasons.

For the period 2015-2017, the regional averages of both RMSE and ERR$_{TOT}$ exhibit, on average, 14% and 11% higher values during the non-growing season compared to the growing season (Figure R2). In contrast, the regional averages of $h\left(err_{f_e}\right)$ and $h\left(err_{f_t}\right)$ have greater values during the growing season by 0.91 [0.85, 0.98] (mean [95% confidence interval]) and 1.29 [1.06, 1.54] ppm compared to the non-growing season (0.67 [0.63, 0.70] and 1.16 [0.94, 1.37] ppm) because of the tendency for errors in terrestrial biosphere $CO_2$ fluxes to increase proportionally with the magnitude of flux values. Consequently, transport errors, inferred from the difference between RMSE and $h\left(err_{f_t}\right)$, are greater in the non-growing season. Given the higher net $CO_2$ emissions in East Asia during the non-growing season, when terrestrial biosphere $CO_2$ uptake is less active, this result is consistent with a previous study showing that transport errors are proportional to the magnitude of the net $CO_2$ flux (Schuh et al., 2019). In addition, we found that the ratio of $h\left(err_{f_t}\right)$ to $h\left(err_{f_e}\right)$ is slightly lower during the non-growing season with 0.58 [0.44, 0.72] compared to the growing season with 0.70 [0.53, 0.89], indicating a relatively greater underestimation of true flux errors when the contributions of anthropogenic $CO_2$ emissions to atmospheric $CO_2$ changes are higher. This supports our finding that the current inverse model may exhibit a systematic bias related to anthropogenic emissions. Furthermore, the consistent ratio of $h\left(err_{f_t}\right)$ to $h\left(err_{f_e}\right)$ below 1, without statistically significant seasonal variations in East Asia, suggests that our conclusions, drawn from the analysis of seven regions, are not seasonally dependent.

[Figure]

**Figure R1** Number of months selected as the effective area for airborne measurements in East Asia during (a) the vegetation growing season and (b) the non-growing season for the period 2015–2017.

[Figure]

**Figure R2** Mean values of monthly (a) RMSE, ERR$_{MIP}$, ERR$_{OBS}$, (b) Ratio, (c) $h(err_{f_e})$, and $h(err_{f_t})$ in East Asia during each season for the period 2015–2017. The error bars represent the 95% confidence intervals derived from 1000 bootstrap samples of datasets.

Next, in order to capture the signals from the surface $CO_2$ fluxes and include as much observation data as possible in our analysis, we used atmospheric $CO_2$ data observed and simulated within the range of 1-5km altitude (above the ground). To assess the sensitivity of our results to the choice of altitude range, we compared the error quantities derived from atmospheric $CO_2$ data within two altitude (above the ground) ranges: 1-3 km and 1-5 km. Among the seven analyzed regions, Australia and South America were excluded in this additional analysis due to having fewer than 100 total observed grid points for the analysis period and losing over 30% of the grid points when narrowing the altitude range. This exclusion was necessary as it could substantially alter the areas represented by our error statistics.

The areas sensitive to airborne $CO_2$ measurements within the two altitude ranges exhibit nearly identical spatial patterns in Alaska, Mid-latitude Norther America, Europe, East Asia, and Southeast Asia, indicating that observations at lower altitudes are more sensitive to surface $CO_2$ fluxes (Figure R3). Because of the higher sensitivity, error statistics in all regions have larger values when calculated using data from the 1-3 km altitude range compared to the 1-5 km altitude range (Figure R4). For example, in Mid-latitude North America, the regional averages of RMSE, ERR$_{TOT}$, $h(err_{f_e})$, and $h(err_{f_t})$ are 1.42 [1.35, 1.48], 1.34 [1.30, 1.39], 0.73 [0.70, 0.76], and 0.86 [0.72, 1.00] ppm when calculated using data within the 1-3 km altitude range. In comparison, when computed from the data within the 1-5 km altitude range, these values are 1.20 [1.15, 1.25], 1.09 [1.06, 1.13], 0.58 [0.56, 0.60], and 0.77 [0.66, 0.88] ppm. However, the ratio of $h(err_{f_t})$ to $h(err_{f_e})$, which is a key metric for quantifying the true regional terrestrial biosphere flux error, did not show significant differences based on the altitude ranges, with the difference being between 0.02 and 0.1. Again, these results suggest that our observation-based regional flux error estimates are not sensitive to the choice of altitude range.

[Figure]

**Figure R3** Number of months selected as the effective area for airborne measurements made within (a) the 1-5 km altitude range and (b) 1-3 km altitude range in Alaska, Mid-latitude Norther America, Europe, East Asia, and Southeast Asia for the period 2015–2017. The outlined area represents selected areas for more than eight months or equal.

[Figure]

**Figure R4** Mean values of monthly (a) RMSE, ERR$_{MIP}$, ERR$_{OBS}$, (b) Ratio, (c) $h\left(err_{f_e}\right)$, and $h\left(err_{f_t}\right)$ derived from atmospheric $CO_2$ data within either the 1-5 km or 1-3 km altitude range for each region for the period 2015–2017. The error bars represent the 95% confidence intervals derived from 1000 bootstrap samples of datasets.

We have added Figures R1, R2, R3, and R4 to the revised supplementary information and added the above explanation on the discussion part of the revised manuscript as follow:

"The performance of inverse models in simulating atmospheric $CO_2$ may vary by season. Airborne measurements were not uniformly conducted across all seasons in most analyzed regions, necessitating an examination of whether the regional averages of error statistics (e.g., Figures 4h, 4i, and 5h) are significantly different with seasons. Among the seven regions analyzed, in East Asia, the Contrail program has continuously conducted $CO_2$ measurements over three years with routes repeated throughout all seasons. This has resulted in the most sensitive area to the measurements exhibiting similar spatial patterns in the vegetation growing season (from May to October) and non-growing season, encompassing the northeast part of China, the Korean Peninsula, and Japan (Figure S6). The airborne measurements in East Asia offer a unique opportunity to isolate the seasonal impacts on regional error statistics. For the period of 2015–2017, the regional averages of both RMSE and ERR$_{TOT}$ exhibit, on average, 14% and 11% higher values during the non-growing season compared to the growing season (Figure S7). In contrast, the regional averages of $h\left(err_{f_e}\right)$ and $h\left(err_{f_t}\right)$ have greater values during the growing season by 0.91 [0.85, 0.98] and 1.29 [1.06, 1.54] ppm compared to the non-growing season (0.67 [0.63, 0.70] and 1.16 [0.94, 1.37] ppm) because of the tendency for errors in terrestrial biosphere $CO_2$ fluxes to increase proportionally with the magnitude of flux values. Consequently, transport errors, inferred from the difference between RMSE and $h\left(err_{f_t}\right)$, are greater in the non-growing season. Given the higher net $CO_2$ emissions in East Asia during the non-growing season, when terrestrial biosphere $CO_2$ uptake is less active, this result is consistent with a previous study showing that transport errors are proportional to the magnitude of the net $CO_2$ flux (Schuh et al., 2019). In addition, we found that the ratio of $h\left(err_{f_t}\right)$ to $h\left(err_{f_e}\right)$ is slightly lower during the non-growing

season with 0.58 [0.44, 0.72] compared to the growing season with 0.70 [0.53, 0.89], indicating a relatively greater underestimation of true flux errors when the contributions of anthropogenic $CO_2$ emissions to atmospheric $CO_2$ changes are higher. This supports our finding that the current inverse model may exhibit a systematic bias related to anthropogenic emissions. Furthermore, the consistent ratio of $h(err_{f_t})$ to $h(err_{f_e})$ below 1, without statistically significant seasonal variations in East Asia, indicates that our conclusions, drawn from the analysis of seven regions, are not seasonally dependent.

To capture the signals from surface $CO_2$ fluxes and maximize observation data in our analysis, we used atmospheric $CO_2$ data observed and simulated within the 1-5 km altitude range. The choice of altitude range may influence regional error statistics, as the performance of inverse models varies with altitude. To gauge this sensitivity, we compared error statistics derived from atmospheric $CO_2$ data with two altitude (above the ground) ranges: 1-3 km and 1-5 km. Among the seven analyzed regions, Australia and South America were excluded in this additional analysis due to having fewer than 100 total observed grid points for the analysis period and losing over 30% of the grid points when narrowing the altitude range. The areas sensitive to airborne $CO_2$ measurements within the two altitude ranges exhibit nearly identical spatial patterns in Alaska, Mid-latitude Norther America, Europe, East Asia, and Southeast Asia, indicating that observations at lower altitudes are more sensitive to surface $CO_2$ fluxes (Figure S8). Because of the higher sensitivity, error statistics in all regions have larger values when calculated using data from the 1-3 km altitude range compared to the 1-5 km altitude range (Figure S9). For example, in Mid-latitude North America, the regional averages of RMSE, ERR$_{TOT}$, $h(err_{f_e})$, and $h(err_{f_t})$ are 1.42 [1.35, 1.48], 1.34 [1.30, 1.39], 0.73 [0.70, 0.76], and 0.86 [0.72, 1.00] ppm when calculated using data within the 1-3 km altitude range. In comparison, when computed from the data within the 1-5 km altitude range, these values are 1.20 [1.15, 1.25], 1.09 [1.06, 1.13], 0.58 [0.56, 0.60], and 0.77 [0.66, 0.88] ppm. However, the ratio of $h(err_{f_t})$ to $h(err_{f_e})$, which is a key metric for quantifying the true regional terrestrial biosphere flux error, did not show significant differences based on the altitude ranges, with the difference being between 0.02 and 0.1. Again, these results suggest that our observation-based regional flux error estimates are not sensitive to the choice of altitude range."

3. Lines 124-125: "measurements made between 1 and 5 km altitude" does not specify whether this means above ground level or above sea level. This needs to be specified. Furthermore, if this altitude range is above sea level then it is entirely possible that highly-variable PBL measurement data are included in the evaluation data, since many aircraft data were collected over topography with surface elevations of hundreds of meters ASL. This would cloud the analysis with noisy measurements having strong signals of local exchange.

[Reply] We really appreciate your comment. It turns out that our previous analysis was based on atmospheric CO2 data within the 1-5 km altitude range above sea level, not ground level. We re-calculated all our results using the atmospheric $CO_2$ data within 1-5 km altitude range "above ground level". The newly computed results, particularly the ratio of RMSE and ERR$_{TOT}$ and the ratio of $h(err_{f_t})$ to $h(err_{f_e})$, key metrics for assessing and quantifying regional terrestrial biosphere flux errors, do not exhibit significant differences compared to the previous results (Table R2).

**Table R2** Mean values of the regionally averaged ratios of RMSE to ERR$_{TOT}$ and the ratios of $h(err_{f_t})$ to $h(err_{f_e})$ for 2015–2017 with their 95% confidence intervals derived from 1000 bootstrap samples of datasets, calculated using

atmospheric $CO_2$ datasets within the range of 1-5 km altitude above sea level (previous results) or above ground level (revised results).

| | | Alaska | Mid NA | Europe | East Asia | Southeast Asia | Australia | South America |
|---|---|---|---|---|---|---|---|---|
| Previous results | RMSE/ERR$_{TOT}$ | 1.04 [0.93, 1.14] | 0.90 [0.83, 0.97] | 0.79 [0.62, 0.98] | 0.84 [0.78, 0.91] | 0.75 [0.66, 0.85] | 0.76 [0.61, 0.90] | 1.01 [0.81, 1.25] |
| | $h(err_{f_t})/h(err_{f_e})$ | 1.07 [0.84, 1.31] | 0.73 [0.59, 0.86] | 0.52 [0.29, 0.77] | 0.59 [0.48, 0.70] | 0.56 [0.42, 0.71] | 0.61 [0.35, 0.91] | 1.03 [0.49, 1.59] |
| Revised results | RMSE/ERR$_{TOT}$ | **0.98 [0.89, 1.08]** | **0.91 [0.84, 0.97]** | **0.79 [0.61, 0.97]** | **0.87 [0.81, 0.94]** | **0.75 [0.65, 0.86]** | **0.73 [0.59, 0.87]** | **1.03 [0.83, 1.28]** |
| | $h(err_{f_t})/h(err_{f_e})$ | **0.96 [0.76, 1.17]** | **0.75 [0.61, 0.90]** | **0.52 [0.28, 0.78]** | **0.64 [0.53, 0.77]** | **0.56 [0.41, 0.72]** | **0.59 [0.34, 0.87]** | **1.10 [0.51, 1.79]** |

We have clearly addressed that we used atmospheric $CO_2$ data within 1-5 km altitude range "above ground level" for the analysis in the revised manuscript as follow:

"To minimize the influence of local sources and maximize the influence of regional fluxes, we excluded surface measurements and only considered airborne measurements made between 1 and 5 km altitude **above ground level.**"

4. It is not clear whether the analysis excludes measurements that were assimilated in the LNLGIS experiment. This is a fundamental piece of information needed to understand the analysis and should absolutely be explicitly stated. If assimilation data are included, then the entire analysis needs to be considered differently.

[Reply] In this study, only airborne $CO_2$ measurement data not assimilated in the LNLGIS experiment were utilized for analysis. We have clearly addressed this in the revised manuscript as follow:

"In addition, only airborne measurement data **not assimilated in the LNLGIS experiment were used** for analysis."

5. The INPE PFP used in this study data have not been screened for water vapor contamination. This is a known problem with PFPs in humid environments and can lead to both a low bias and spurious variability in CO2 measurements. This is a particular concern with tropical aircraft samples due to expected high humidity of sampled air. There are indications that water vapor contamination can persist in PFP flasks so that even dry high-altitude samples may be affected. This water vapor issue in aircraft PFPs has been documented in Baier et al. (2019, https://agupubs.onlinelibrary.wiley.com/doi/10.1029/2019JD031339) and reported at various meetings (e.g. https://gml.noaa.gov/publications/annual_meetings/2019/abstracts/74-190401-B.pdf). As reported to the authors in OCO-2 meetings, about one-third of historical NOAA PFP measurements have been flagged due to suspected water vapor contamination. In the same meetings the authors were cautioned about this issue affecting INPE PFP data. In ObsPack products, INPE PFP data are all flagged as "do not assimilate", indicating that they are neither suitable for assimilation nor for evaluation purposes. Finally, these data are distributed in a special ObsPack product labeled "restricted" in part to warn users about the problem.

[Reply] We appreciate your notification regarding the water vapor contamination issue in the INPE PFP data. We deliberated whether to include this data in our analysis due to the mentioned issue. However, we decided to include it in our analysis because we believe this study can provide valuable insights for future research utilizing bias-corrected observations to quantify biosphere flux errors in South America, one of the most critical regions for terrestrial carbon cycle studies. To ensure that readers are well-informed about this issue before interpreting our

results, we have clearly addressed the unresolved water vapor contamination issue of this data and its potential impact on our findings in the revised manuscript as follow:

"This dataset includes five flask measurements provided by the National Institute for Space Research (INPE), which might have a higher measurement error due to water vapor contamination compared to other datasets (Baier et al., 2019). Despite their potential limitation, our analysis, aimed at introducing a method for quantifying flux errors, incorporates INPE data to offer guidance for future studies leveraging bias-corrected observations from this region, is critical for terrestrial carbon cycle studies. Readers should keep in mind that our results from South America may have relatively lower reliability compared to that from other regions."

"However, in reality, systematic errors could be present in airborne observation data stemming from instrument or setup biases, calibration offsets, and other factors. Especially, $CO_2$ measurements in South America from INPE might exhibit a higher measurement error compared to other regions because of unresolved water vapor contamination issues in the flask measurements, which could result in both a low bias (0.1 and 0.8 ppm at 1.75% and 3–4% absolute humidity, respectively) and spurious variability (Baier et al., 2019). The potential effects of these systematic errors on our findings will be addressed in Section 4."

"First, due to a lack of information, we set observation measurement errors under ideal conditions. In reality, inadequate quality control can result in significant systematic bias for specific regions and time periods (Masarie et al., 2011; Baier et al., 2019), impacting our results, especially in South America. For instance, if this led to an average measurement error of 0.5 ppm during the analysis period, the calculated true flux error would decrease from 351 to 277 TgC year$^{-1}$ for South America and from 371 to 260 TgC year$^{-1}$ for mid-latitude North America."

6. The CO2 measurement data used in this study have not been correctly cited. It also is not clear whether ObsPack data providers have been properly acknowledged. The OCO-2 ObsPack product is a "composite" product created from seven source ObsPacks. The source products need to be cited following the instructions at https://gml.noaa.gov/ccgg/obspack/citation.php (available also in the distributed metadata). Use of an ObsPack product also includes usage terms which suggest that it may be appropriate to offer coauthorship to the data providers. The seven source ObsPacks are listed in the metadata directory of the downloaded product. In the current draft, only the obspack_co2_1_GLOBALVIEWplus_v6.1_2021-03-01 product is cited, whereas apparently there are data used from five other ObsPacks: the NRT product, the Manaus product, the INPE product, the CONTRAIL product, and the AirCore product.

[Reply] We appreciate your guidance for properly acknowledging the ObsPack products. We have included citation information and the DOI for all types of ObsPack data in Table 1 of the revised manuscript (shown in Table R1). The OCO-2 ObsPack products, we used, are originated from following five different Obspack data: obspack_co2_1_GLOBALVIEWplus_v6.1_2021-03-01, obspack_co2_1_AirCore_v4.0_2020-12-28, obspack_co2_1_INPE_RESTRICTED_v2.0_2018-11-13, obspack_co2_1_NRT_v6.1.1_2021-05-17, obspack_multi-species_1_manaus_profiles_v1.0_2021-05-20.

We also reached out to all airborne $CO_2$ measurement data providers and sought their guidance on proper acknowledgment or co-authorship for utilizing the airborne measurements dataset in this research before submitting this manuscript. During the revision process, SGP flask $CO_2$ measurement data has been included, leading to the invitation of "Sébastien C. Biraud" as a co-author.

**References**

[revised manuscript text omitted]

---

## Author Response (AR1)

**Referee#1**

Yun et al have written an elegant and thought-provoking paper on error quantification for the OCO-2 MIP v10 dataset. The approach is a mixture of theory, where possible, and approximations, otherwise. It is certainly an interesting contribution, but its crude assumptions limit its scope to a scenario ("if we could assume that…, then we could conclude that…"). In fact, in practice, the main result seems to be the inference of... one element of the MIP protocol, namely the fact that fossil fuel fluxes were imposed on all inverse modeling systems. I can only encourage the authors to thoroughly revise their text in order to give it the right perspective.

[Response] We appreciate your constructive comments on this manuscript. We revised the manuscript to fully address your comments and suggestions. Detailed point-by-point responses to your comments and related revisions are presented below. The original comments are in black, and our responses are in blue color.

**Detailed comment:**

- Title: the flux errors here refer to the average of the flux ensemble, not to the individual flux sets. Please correct

  [Response] Based on the reviewer's suggestions, we changed the title as follow:
  "Quantification of regional terrestrial biosphere CO2 flux errors in v10 OCO-2 MIP **ensemble** using airborne measurements"

- 27: this main conclusion is not a finding but an input to the study (l. 118).

  [Response] The main finding of this study is that "actual errors in ensemble mean terrestrial flux estimates from v10 OCO-2 MIP are underestimated in regions with higher fossil fuel emissions compared to terrestrial biosphere fluxes." However, in the original main text, we did not provide information on the anthropogenic emissions in our analysis regions. In the revision, we added fossil fuel emission information to support our findings in the result section and Figure 6 as follow:

  "We find that the actual terrestrial biosphere flux errors are underestimated, particularly in regions where annual $CO_2$ emissions from fossil fuel combustion exceed annual terrestrial biosphere fluxes by 3-31 times. The airborne measurements carried out in mid-latitude North America, East Asia, and Southeast Asia are influenced by a broad region encompassing the United States, the eastern part of East Asia, and the western part of Southeast Asia where fossil fuel $CO_2$ emissions are 1,341, 2,443, and 815 Tg C year$^{-1}$, respectively. The first two regions are estimated as significant terrestrial biosphere $CO_2$ sinks, with estimated fluxes of −414 ± 279 (ensemble mean ± 1σ) and −561 ± 380 Tg C year$^{-1}$, in contrast to Southeast Asia (26 ± 118 Tg C year$^{-1}$). However, the $CO_2$ sinks are more than 3 and 4 times smaller than the fossil fuel $CO_2$ emissions, respectively. The recalculated terrestrial biosphere flux errors in these regions exceed the ensemble spread with values of 374, 643, and 211 Tg C year$^{-1}$. Observations in Europe and Australia, conducted over limited periods and specific locations, mainly represent certain areas in the western Europe and the southeastern part of Australia, where fossil fuel emissions (234 and 53 Tg C year$^{-1}$, respectively) are around four and five times greater than terrestrial biosphere sinks (−51 ± 34 and −10 ± 67 Tg C year$^{-1}$). The recalculated terrestrial biosphere flux errors in these regions are also larger than the ensemble spread, estimated at 65 and 114 Tg C year$^{-1}$, respectively. On the contrary, the most influential areas for the observation in Alaska and South America, encompassing the southeastern region of Alaska and the northern part of Brazil, characterized as a terrestrial biosphere sinks of −8 ± 11 Tg C year$^{-1}$ and sources of 625 ± 387 Tg C year$^{-1}$, respectively, which are comparable to or more than 10 times greater than fossil fuel emissions (10 and 38 Tg C year$^{-1}$). The observation-based estimates of true terrestrial biosphere

flux errors are almost identical to the ensemble spread in both regions with values of 11 and 398 Tg C year$^{-1}$, respectively."

[Figure]

**Figure 6:** (a) Number of months selected as the effective area for airborne measurements. The outlined area represents selected areas for more than eight months or equal. (b) Annual total terrestrial biosphere $CO_2$ fluxes obtained from the ensemble mean of ten OCO-2 MIP models and annual total fossil fuel $CO_2$ emissions estimated from ODIAC data for each outlined area averaged over the period 2015–2017. The error bars in black and red indicate the one standard deviation of the inversion estimates and the newly estimated error range from this study, respectively.

Moreover, this study does not specify major reasons for the underestimation in true flux errors, but discusses possible causes for the underestimation. The one element of MIP protocol, "OCO-2 MIP models treat the fossil fuel emissions as true values and use the same dataset", could be one possible explanation for our findings but we did not (cannot) conclude that it is the main source of the underestimation in the true flux errors. This study aims to develop a framework to quantify the errors in regional terrestrial biosphere $CO_2$ fluxes estimated from an ensemble of inverse models. Evaluating impacts of each possible error source on ensemble flux estimates is out of the scope of this study. To better convey our main findings, we revised the corresponding sentences in the abstract as follows: "By identifying the most sensitive areas to airborne measurements through adjoint sensitivity analysis, we find that the underestimation of biosphere flux errors is prominent in eastern parts of Australia and East Asia, western parts of Europe and Southeast Asia, and midlatitude North America where the magnitudes of annual fossil fuel emissions exceed those of annual biosphere fluxes by 3-31 times over the three years. The regions with no underestimation are southeastern Alaska and northeastern South America where fossil fuel emissions are comparable to or less than biosphere fluxes." We also adjusted related statements in the main text.

- 53: RECCAP seems to be driven by GCP (https://www.globalcarbonproject.org/reccap/overview.htm) – what is the difference with the previous item (l. 52)?

  [Response] We agree that RECCAP activities utilizes the results from GCP. We revise that sentence as follow:
  "These projects include the TransCom project (Gurney et al., 2004; Houweling et al., 2015), which was first initiated in 1990s, as well as subsequent projects such as **the Global Carbon Project** (GCP; Friedlingstein et al., 2023; Ciais et al., 2022) and the Orbiting Carbon Observatory-2 (OCO-2) MIP (Crowell et al., 2019; Peiro et al., 2022; Byrne et al., 2023)."

- 66: please insert "explicitly" before "incorporate", as the difference between systematic errors and error correlations can be subtle

  [Response] We revise that sentence as follow:

  "This Bayesian posterior uncertainty accounts for random errors in the prior fluxes and observations but does not **explicitly** incorporate systematic errors, thus providing a potential underestimate of the total posterior error."

  77: it would be fairer to write that this method has no theoretical basis. "lacks" may suggest that there is hope to find one (or, please, elaborate).

  [Response] We revise that sentence as follow:
  "… but this method **does not have** an observational and theoretical basis and may not reflect actual errors"

- 93: the given definition of "error" is surprising, because "observed" is vague (by which technique?), and because the previous sentence is about fluxes.

  [Response] We revise that sentence as follow:
  "Here, "error" refers to the magnitude of the differences between **the true and estimated flux values**, without considering the sign."

- 116: ten members only, covering only four transport models. How can their statistics be robust? Briefly touched in l. 483-7, but too late.

  [Response] Our estimates of flux and transport errors are computed from the ensemble spread of posterior $CO_2$ concentrations. Since the ensemble members encompass only four types of transport models, the estimated transport errors may not fully capture the actual transport errors. Thus, not only the disparity between the estimates and actual flux errors but also the discrepancy between the actual transport errors and their estimates could contribute to the differences between $ERR^2_{TOT}$ and $RMSE^2$. Therefore, we added the following discussions in the method and discussion sections in the revised manuscript, respectively:

  "However, our assumption regarding transport errors may be a strong assumption given that the transport errors are derived from 10 ensemble members, covering four different transport models, which might not fully capture the actual transport errors. We discuss how this assumption affects our key results in Section 4."

  "We further investigate how our main results would be affected if the estimated transport errors deviate from actual errors by 20% and 40% of the difference between $RMSE^2$ and $ERR^2_{TOT}$. The ratio of regional mean of $h\left(err_{f_e}\right)$ to $h\left(err_{f_t}\right)$ increases by, on average, only up to 0.04 and 0.09 in the seven regions throughout the analysis period, respectively (Figure S11). In both cases, the estimated flux errors in mid-latitude North America, Europe, East Asia, and Southeast Asia still show significant underestimation at a 95% confidence level, while not in Alaska and South America. In Australia, characterized by a wide

uncertainty range, significant underestimation is also observed in the 20% cases, supporting the robustness of our findings. In the future OCO-2 MIP, the participation of inverse modeling groups using other transport models or meteorological forcing data might contribute to estimating transport errors closer to actual values."

[Figure]

**Figure S11.** Mean values of monthly $h\left(err_{f_e}\right)$, and $h\left(err_{f_t}\right)$ for each region for the period 2015–2017 where (a) $h\left(err_{f_t}\right)^2 - h\left(err_{f_e}\right)^2 = 0.8 \cdot (RMSE^2 - ERR_{TOT}^2)$ and (b) $h\left(err_{f_t}\right)^2 - h\left(err_{f_e}\right)^2 = 0.6 \cdot (RMSE^2 - ERR_{TOT}^2)$. The numbers at the top of each panel indicate the ratio of the three-year mean $h\left(err_{f_e}\right)$ to $h\left(err_{f_t}\right)$. The error bars and error ranges represent the 95% confidence intervals derived from 1000 bootstrap samples of datasets.

- 148: please insert "below" before "to evaluate"

[Response] We revise that sentence as follow:
"We first employed the two matrixes defined in Eq. (1) and (2) **below** to evaluate ensemble posterior flux errors proposed by Liu et al. (2021)."

- 160: the way this exclusion is done biases the statistics towards the model values. Awkward.

[Response] We agree on the reviewer's comment. We eliminated this process which excludes outliers and re-calculated all error quantities by using all observation data in the analysis. Since outliers comprised 0.05% of the total data, the newly computed results, particularly the ratio of $ERR_{TOT}$ and RMSE and the ratio of $h\left(err_{f_e}\right)$ to $h\left(err_{f_t}\right)$, key metrics for assessing and quantifying regional terrestrial biosphere flux errors, do not exhibit significant differences compared to the previous results (Table R1).

**Table R1** Mean values of the regionally averaged ratios of $ERR_{TOT}$ to RMSE and the ratios of $h\left(err_{f_e}\right)$ to $h\left(err_{f_t}\right)$ for 2015–2017 with their 95% confidence intervals derived from 1000 bootstrap samples of datasets, calculated using atmospheric $CO_2$ datasets within the range of 1-5 km above ground level when excluding (previous results) or including outliers (revised results).

| | | Alaska | Mid NA | Europe | East Asia | Southeast Asia | Australia | South America |
|---|---|---|---|---|---|---|---|---|
| | $ERR_{TOT}$/RMSE | 0.98 [0.89, 1.08] | 0.91 [0.84, 0.97] | 0.79 [0.61, 0.97] | 0.87 [0.81, 0.94] | 0.75 [0.65, 0.86] | 0.73 [0.59, 0.87] | 1.03 [0.83, 1.28] |

| | | | | | | | | |
|---|---|---|---|---|---|---|---|---|
| Previous results | $h(err_{f_e})/h(err_{f_t})$ | 0.96 [0.76, 1.17] | 0.75 [0.61, 0.90] | 0.52 [0.28, 0.78] | 0.64 [0.53, 0.77] | 0.56 [0.41, 0.72] | 0.59 [0.34, 0.87] | 1.10 [0.51, 1.79] |
| Revised results | $ERR_{TOT}/RMSE$ | 0.98 [0.89, 1.08] | 0.90 [0.83, 0.97] | 0.79 [0.61, 0.97] | 0.84 [0.78, 0.91] | 0.75 [0.65, 0.86] | 0.73 [0.59, 0.87] | 0.99 [0.79, 1.24] |
| | $h(err_{f_e})/h(err_{f_t})$ | 0.96 [0.76, 1.17] | 0.74 [0.61, 0.88] | 0.52 [0.27, 0.78] | 0.59 [0.48, 0.70] | 0.56 [0.41, 0.72] | 0.59 [0.34, 0.87] | 0.97 [0.49, 1.54] |

- 180-1: strong approximation. Also, the fact that the simulations were made at half-degree resolution, which is so much coarser than the measurements

[Response] Before addressing this comment, we'd like to mention that we changed the notation of representation error estimates from $ERR_{o\_r}$ to $ERR_{REP}$ because in the revised version, we separately treat representation errors and observation (measurement) errors to convey our approach more clearly.

Our results show that the regional mean representation errors ($ERR_{REP}$) have lower monthly variability (i.e., standard deviation) ranging from 0.12 to 0.24 ppm compared to the variability of RMSE (0.24 to 0.45 ppm) and $ERR_{MIP}$ (0.18 to 0.45 ppm) across all regions, except for South America, where the observational data are sparse (Figure 4). In addition, considering that we repeatedly used the high-resolution GEOS-Chem results for 2018 in the $ERR_{REP}$ calculation for 2015-2017, we also examined the monthly variability of $ERR_{REP}$ on an annual basis, focusing on the regions with year-round aircraft observation data. We found that their range is just 0.12-0.19 ppm and 0.13-0.20 ppm in North America and East Asia during 2015-2017, respectively, and 0.14-0.18 ppm over Southeast Asia during 2015-2016. Given that seasonal changes in atmospheric circulations and surface $CO_2$ fluxes are main drivers of the spatiotemporal variations of atmospheric $CO_2$ (Umezawa et al., 2018), the interannual variability of monthly CO2 variances within 2°x2.5° grid cells ($VAR_{CO_2}$) is anticipated to be lower than the sub-annual variability. Therefore, using the GEOS-Chem results from the same year repeatedly or from corresponding years for $ERR_{REP}$ calculation would not lead to significant differences.

We revised the sentence by adding rationale for our assumption as follows:
"It is assumed that the variances do not vary significantly across years, **given relatively lower monthly variability of $ERR_{REP}$ compared to that of RMSE and $ERR_{MIP}$ (to be shown in Section 3.2).**"

Next, considering the disparity in spatial coverage between the GEOS-Chem model grid (0.5-degree) and observation points, we calculated the mean representation errors at a 1-degree grid scale and utilized it in our study. We agree on the importance of validating whether our representation errors, derived from simulated atmospheric $CO_2$ fields with 0.5-degree resolution, reasonably represents the actual spatial variability of $CO_2$ concentration within a 1-degree grid. For the validation, we used aircraft measurements from ACT-America project which provides extensive atmospheric $CO_2$ data across central and eastern North America spanning nine months for the period 2016-2019. Airborne observations do not provide simultaneous spatial distribution information of $CO_2$, unlike models. Thus, we calculated the variances of observed $CO_2$ concentrations within each 1°x1° grid cell with a 500 m vertical interval (from 1 to 5 km) at 3-hour intervals ($OBS\_ERR_{REP}^2$). On average, each grid box includes around 70 aircraft observation data. For comparison with the $OBS\_ERR_{REP}^2$, we sampled $VAR_{CO_2}$, derived from GEOS-Chem results, at the corresponding aircraft measurement times and locations at each grid box and computed their mean values ($MOD\_ERR_{REP}^2$). Monthly and regional mean $OBS\_ERR_{REP}$ and $MOD\_ERR_{REP}$ over North America show a significant positive correlation (r= 0.72, p <0.05) for the ACT-America project period (Figure S1). $MOD\_ERR_{REP}$ also has a similar mean value (0.62 [0.59, 0.64] ppm) with that of $OBS\_ERR_{REP}$ (0.49 [0.47, 0.51] ppm). This result supports that $ERR_{REP}$, based on 0.5-degree Geos Chem simulation results, could reasonably represent the actual mean observation representation error at a 1-degree grid scale.

[Figure]

**Figure S1** Monthly variations of $OBS\_ERR_{REP}$ and $MOD\_ERR_{REP}$ in the central and eastern North America for ACT-America project period during 2016–2019. The lines and shaded areas represent the mean and the 95% confidence intervals derived from 1000 bootstrap samples of the 1-degree grid-based monthly error quantities.

We added Figure S1 and above explanation on the supplementary information.

Umezawa, T., Matsueda, H., Sawa, Y., Niwa, Y., Machida, T., and Zhou, L.: Seasonal evaluation of tropospheric $CO_2$ over the Asia-Pacific region observed by the CONTRAIL commercial airliner measurements, Atmospheric Chemistry and Physics, 18, 14851–14866, https://doi.org/10.5194/acp-18-14851-2018, 2018.

197-198: This method mainly relies on this approximation, but I see no justification. You need to convince the reader that it is reasonable. Based, e.g., on Schuh et al (2019), "The research suggests that variability among transport models remains the largest source of uncertainty across global flux inversion systems", cited in l. 49, I would be surprised if it was, but please explain why I am wrong!

[Response] Our assumption is not in conflict with previous studies (Schuh et al., 2019) suggesting that transport errors are one of major sources of uncertainty in current global inversion estimates because transport errors are accounted for in both $RMSE^2$ and $ERR_{TOT}^2$. True transport errors are incorporated within $RMSE^2$ and estimated transport errors, computed from the ensemble spread among transport models used in OCO-2 MIP, are included in $ERR_{TOT}^2$. In this study, we assume that the estimated transport errors represent the actual transport errors. Thus, given that our estimates of representation errors reasonably depict actual representation errors, the difference between $RMSE^2$ and $ERR_{TOT}^2$ mainly arises from the difference in the flux error variances ($\sigma_{f_t}^2$ and $\sigma_{f_e}^2$). However, we agree that our assumption regarding transport errors may be a strong assumption given that the transport errors are derived from 10 ensemble members, covering four different transport models, which might not fully capture the actual transport errors. We added discussions in the revised manuscript regarding the potential impact of errors in this assumption on our main findings as we addressed in our response to the reviewer's comment on L116. We also revised sentences to clarify the assumption applied in this study (i.e., lines 236-240 and 267-269).

- Figure 1, point 1): is actually about flux+transport errors

[Response] We agree with the reviewer's comment that the ratio of $ERR_{TOT}^2$ to $RMSE^2$ indicates whether posterior flux and transport errors computed from the ensemble spread overestimate or underestimate

true errors in the ensemble mean of posterior fluxes and transport. However, as addressed in our response to the reviewer's previous comment on L197-198, this study assumes that the estimated transport errors from the ensemble spread among transport models used in OCO-2 MIP represent actual transport errors. To effectively convey our perspective, we added these sentences to the revised manuscript:

"**Given that $ERR_{REP}^2$ reasonably depict actual representation errors, $Ratio^2$ can indicate whether posterior flux and transport errors computed from the ensemble spread is an overestimation or underestimation of true flux and transport errors. In this study, we assume that the estimated transport errors from the ensemble spread among transport models used in OCO-2 MIP represent the true transport errors and the difference between $RMSE^2$ and $ERR_{TOT}^2$ mainly arises from the difference in the flux error variances ($\sigma_{f_t}^2$ and $\sigma_{f_e}^2$).** Thus, a ratio close to 1 indicates that the estimated posterior flux errors derived from the ensemble model spread are close to the true posterior flux error in the ensemble mean fluxes. A ratio greater than 1 means that the posterior flux errors are overestimated, and vice versa**.**"

- 215: the concept of forward simulations obtained with a (backward-running) adjoint model is not intuitive. Did you use the adjoint to compute the Jacobian matrix and then did you run it forward?

  [Response] The GEOS-Chem Adjoint model integrates the forward GEOS-Chem and its derivative adjoint codes (https://wiki.seas.harvard.edu/geos-chem/index.php/GEOS-Chem_Adjoint_User%27s_Guide#Brief_overview). Within the model code, users have the option to choose whether to perform only forward simulation, in the same way used in a forward GEOS-Chem model, or to calculate adjoint sensitivity values as well. For this analysis, we choose the forward simulation options and derived simulated $CO_2$ concentration fields from the prescribed surface $CO_2$ fluxes.

  I have revised the sentence as follows to convey the information more clearly.

  "To get $h_{GC}(err_{f_e})^2$, we conduct a set of forward simulations using the GEOS-Chem transport model (within the GEOS-Chem Adjoint model v8.2j; Henze et al., 2007)."

- 220: if I understand it well, sub-monthly patterns are fixed, even though the comparison is to instantaneous measurements. The spread should be largely underestimated.

  [Response] In this study, our objective is to assess errors in the ensemble mean of posterior terrestrial biosphere $CO_2$ fluxes at regional scales on a monthly basis rather than a sub-monthly basis. Accordingly, all error quantities, including $ERR_{TOT}$, $RMSE$ , $h(err_{f_e})$, $h(err_{f_t})$, $err_{f_e}$, and $err_{f_t}$, are computed on a monthly time scale. Thus, it is appropriate to use monthly posterior flux estimates for calculating atmospheric $CO_2$ errors solely attributed to the ensemble spread of monthly posterior fluxes from OCO-2 MIP ($h(err_{f_e})$).

- Eq. (13): my previous comments challenge it

  [Response] Please take a look at our responses to your previous comments.

- 226: again, I do not trust this hypothesis

  [Response] Please take a look at our response to your previous comment on L197-198.

  289: RMSE has already been defined

  [Response] We revised that sentence as follow:

  "…we compared RMSE with the sum of $ERR_{MIP}$, $ERR_{REP}$, and $ERR_{OBS}$ (referred to as $ERR_{TOT}$)."

- 300: Liu et al. (2022) is missing. I am looking for it to read the basis of "indicating most inverse models have common significant errors for this region".

  [Response] Apologies for any confusion. It's actually Liu et al. (2021), not Liu et al. (2022). We revised the citation information properly. Liu et al. (2021) showed an underestimation of posterior flux errors in CMS-Flux inverse model.

  Liu, J., Baskaran, L., Bowman, K., Schimel, D., Bloom, A. A., Parazoo, N. C., Oda, T., Carroll, D., Menemenlis, D., Joiner, J., Commane, R., Daube, B., Gatti, L. V., McKain, K., Miller, J., Stephens, B. B., Sweeney, C., and Wofsy, S.: Carbon Monitoring System Flux Net Biosphere Exchange 2020 (CMS-Flux NBE 2020), Earth System Science Data, 13, 299–330, https://doi.org/10.5194/essd-13-299-2021, 2021.

- 493: based on the above, I would challenge this statement.

  [Response] Please take a look at our responses to your previous comments.

**Referee#2**

Review.

This is an interesting and creative manuscript that I believe makes useful progress toward a challenging and important objective - evaluating uncertainty in the results of inverse estimates of biogenic CO2 fluxes. I believe the results and conclusions are justified given what I can gather from the data presented. My primary concern is the clarity of the text, both the methods and the results. At worst I could not understand some of the methods and results, and in other areas I think that I understand, but the presentation makes understanding a struggle.

I would encourage the authors to consider revising some of the presentation to make this important work more accessible. I have two main concerns.

[Response] We appreciate your constructive comments on this manuscript. We revised the manuscript to fully address your comments and suggestions. Detailed point-by-point responses to your comments and related revisions are presented below.

1. The authors work hard to explain the methods, but I struggled to follow. The figure is a good idea, and the appendix is very helpful. I found, however, that the terminology used, including the mathematical symbols used to define terms, was revealed gradually and irregularly. This makes reading the document difficult. I would strongly recommend presenting the most important variables and their definition up front and early in the text, and making sure to stick to that terminology and variable set throughout. I would make it easy for the reader to quickly look up the meaning of the most important variables used in the main results.

[Response] We appreciate your suggestions. We revised the method section to introduce main error statistics early on in the text. In addition, we revised many sentences to convey our approach more clearly. For detailed information about the revisions we made, please refer to our responses to your detailed comments.

2. Some of the presentation of results needs, in my opinion, to be rewritten. Some of the results are not organized into clearly written paragraphs, with a key finding as the topic sentence and discussion in the paragraph that explains the reasoning behind that key finding. Instead there are paragraphs that tend toward describing the figures, raising conclusions mid-paragraph or at the end of the paragraph, and those conclusions are clearly linked (in my mind) to the preceding text. I believe that rewriting some of the results and discussion (see detailed notes) will make the document easier to follow and more clearly illustrate what appear to be an interesting set of results derived from a creative set of methods.

[Response] Following your suggestions, we significantly revised many paragraphs in the results and discussion sections to ensure that our key findings and messages are effectively conveyed. For detailed information about the revisions we made, please refer to our responses to your detailed comments.

3. I have one question about the content. The number of airborne observations (which is not well defined, see my notes below) vary dramatically from region to region. I would expect this to have a much larger impact on the results than it appears to have. Heavily sampled regions (e.g. N America) don't appear much better understood than severely undersampled regions (e.g. S. America). Should we infer that intensive aircraft campaigns are not very beneficial, and that very limited sampling provides sufficient information for evaluating uncertainties in

inversions? Or that large investments in sampling does not greatly improve our understanding? Or is it safer to say that we have not yet learned how to use extensive data set to our greatest benefit?

[Response] The availability of airborne observation data in each region impacts both the reliability of our error statistics for quantifying regional flux errors and the area extents represented by these statistics (details described in our responses of your 21st and 34th comments). For example, the ratios of three-year mean $h\left(err_{f_e}\right)$ to $h\left(err_{f_t}\right)$, which are key metrics for quantifying regional flux errors (Figure 6h), have a smaller uncertainty range in the mid-latitude North America (0.75 [0.61, 0.89]; mean [95% confidence intervals]) and East Asia (0.59 [0.48, 0.70]), where wide and consistent data coverage are available, than other regions, particularly Europe (0.52 [0.27, 0.78]) and South America (0.97 [0.49, 1.54]), where observation coverage is sparse and intermittent. In addition, our three-year mean error statistics computed from data in mid-latitude North America and East Asia represent broad regions encompassing the United States and eastern parts of East Asia. In contrast, those computed from data in Alaska and Europe, where observation made for limited periods and at specific locations, represent much smaller regions. These results imply that intensive aircraft campaigns are critical for reliable evaluation and quantification of the errors in regional terrestrial flux estimates derived from inverse models. We believe that substantial investments in airborne sampling are undoubtedly beneficial in understanding the sources of errors in current inverse modeling and in estimating terrestrial biosphere $CO_2$ flux more accurately.

We included above explanations in the revised manuscript (lines 275-279, 448-450, and 513-527).

In sum I find the document very much worthy of publication, but in need of work on the presentation.

Detailed comments:

1. Lines 23 and 25. Are these references to fluxes specific to biogenic CO2 fluxes? At a few places in the abstract it isn't clear what fluxes are included. This gets especially confusing on line 27 when anthropogenic CO2 emissions are specified.

[Response] We revised these sentences as follow:

"…the observation-based error estimates exceed the atmospheric $CO_2$ errors computed from the ensemble spread of posterior **biosphere** $CO_2$ flux estimates by 1.33-1.93 times, …. By identifying the most sensitive areas to the airborne measurements through adjoint sensitivity analysis, we find that the underestimation of **biosphere** flux errors is prominent in eastern parts of Australia and East Asia, western parts of Europe and Southeast Asia, and midlatitude North America where the magnitudes of annual fossil fuel emissions exceed those of annual biosphere fluxes **by 3-31 times over the three years**. The regions with no underestimation were southeastern Alaska and northeastern South America **where fossil fuel emissions are comparable to or less than biosphere fluxes.**"

2. Line 36-37. English needs some work.

[Response] We revised these sentences as follow:

"Accurate estimates of regional terrestrial biosphere carbon fluxes and their uncertainties are, therefore, crucial for monitoring changes in terrestrial carbon sinks."

3. Line 40. final phrase is left dangling.

[Response] We revised these sentences as follow:

"Atmospheric $CO_2$ inverse modeling is one of the widely employed approaches to estimate terrestrial and air-sea $CO_2$ fluxes **by assimilating observed atmospheric $CO_2$ concentrations**."

4. Line 48. I'm not sure what "systematic errors in …. inversion setups" means.

[Response] We revised these sentences as follow:

"However, concerns have been raised that the inverse modeling results are **sensitive to the selection of transport models, prior flux datasets, and data assimilation techniques** that are not accounted for in the Bayesian framework."

5. Lines 85-90. This is tough to follow. But let me try the methods, then perhaps this will be clearer.

[Response] We revised these sentences as follow:

"We quantify the errors in ensemble mean estimates of posterior atmospheric $CO_2$ by comparing them with the airborne $CO_2$ data. We then estimate the contributions of various error components (e.g., representation, observation, transport, and flux errors) to the observation-model difference in atmospheric $CO_2$ and isolate the contribution of terrestrial flux errors. Next, we identify the areas that these airborne $CO_2$ are most sensitive to and quantify the annual biosphere flux errors in these areas."

6. Lines 91-92. If the objective focuses on the use of airborne observations, it might help to include some description of these observations and their suitability for this task in the introduction.

[Response] We appreciate the reviewer's suggestion. We include the following sentences in the introduction in the revised manuscript:

"This study uses more than 833,000 airborne CO2 measurement data collected at 1-5 km altitude from 20 different measurement projects (e.g., Baier et al., 2021; Miller et al., 2021; NOAA Carbon Cycle Group ObsPack Team, 2018; Schuldt et al., 2021a; 2021b). These data have broader spatial coverage and are less influenced by local sources compared to surface $CO_2$ data, thus capturing signals from regional surface $CO_2$ fluxes."

7. Line 96. I'm puzzled by the statement, "an approximation of RMSE." Maybe, "RMSE in the elements of the ensemble"? I'm not sure that is clearer.

[Response] The previous expression was not clear, so we revised these sentences as follows:
"First, we define two quantities: 1) the root mean square errors ($RMSE$) between the ensemble mean of posterior $CO_2$ concentrations and observed $CO_2$ concentrations, and 2) $ERR_{TOT}$ (Section 2.3). $RMSE^2$ represents the true errors in OCO-2 MIP ensemble mean of $CO_2$ concentrations including representation errors ($\sigma_r^2$), observation errors ($\sigma_o^2$), true flux errors projected onto $CO_2$ concentration ($\sigma_{f_t}^2$), transport errors ($\sigma_t^2$), and error covariances between the preceding two terms ($cov(\sigma_{f_t}, \sigma_t)$). $ERR_{TOT}^2$ is the sum of the estimated error components, defined as the sum of $ERR_{REP}^2$, $ERR_{OBS}^2$ and $ERR_{MIP}^2$. $ERR_{REP}^2$ and $ERR_{OBS}^2$ indicate representation errors ($\sigma_r^2$) and observation errors ($\sigma_o^2$), respectively. $ERR_{MIP}^2$ is the sum of estimated flux errors projected onto $CO_2$ space ($\sigma_{f_e}^2$) and transport errors ($\sigma_t^2$), and their error covariances ($cov(\sigma_{f_e}, \sigma_t)$), computed from an ensemble spread of posterior $CO_2$ concentrations."

8. Line 99. Next? Did you just present this as (2) in line 96?

[Response] In previous manuscript, the analysis described in line 99-100 differs from that described in (2) in line 96. In line 96, we defined $ERR_{TOT}^2$ (=$ERR_{REP}^2$ + $ERR_{OBS}^2$ + $ERR_{MIP}^2$), where $ERR_{MIP}^2$ indicates the sum of estimated flux errors projected onto $CO_2$ space and transport errors, and their error covariances. $ERR_{MIP}^2$ is computed from an ensemble spread of "posterior CO2 concentrations". However, in the analysis described in line 99-100, we derived atmospheric $CO_2$ errors due to only the ensemble spread of "posterior CO2 flux estimates" ($h(err_{f_e})$) through transport model simulations. We revised these sentences to convey our approach more clearly as follow:

" … $ERR_{MIP}^2$ is the sum of estimated flux errors projected onto $CO_2$ space ($\sigma_{f_e}^2$) and transport errors ($\sigma_t^2$), and their error covariances ($cov(\sigma_{f_e}, \sigma_t)$), computed from an ensemble spread of posterior $CO_2$ concentrations. … Next, we calculate the estimated flux errors projected onto atmospheric $CO_2$ ($h(err_{f_e})$) through atmospheric transport simulations (Section 2.4)."

9. Line 101. What are the true errors? Does this differ from the ratio exercise described earlier?

[Response] RMSE represents the true errors in OCO-2 MIP ensemble mean of $CO_2$ concentrations including true flux errors projected onto $CO_2$ concentration and transport errors, error covariances between the preceding two terms, representation errors, and observation errors. In line 101 of the previous manuscript, the true errors indicate the true errors in ensemble mean posterior fluxes projected onto $CO_2$ space that are included in the RMSE.

In addition, earlier analysis based on the ratio between $ERR_{TOT}$ and RMSE is for evaluating whether the flux errors computed from ensemble spread of posterior flux estimates overestimate or underestimate the true errors in ensemble mean of flux estimates of OCO-2 MIP models. However, in the later analysis, we quantify the true flux errors projected onto $CO_2$ space by isolating them within the RMSE.

We revise that paragraph to convey our approach more clearly (lines 96–114 in the revised manuscript; or please refer to the response in the following comment)

10. Figure 1.  This is a nice idea, but the terms in this figure need to be defined.  At present these terms don't match the terms in the text, and there are many undefined terms in the figure.

[Response] We appreciate the reviewer's suggestion. We revise that paragraph to match with the terms included in Figure 1. In addition, we changed the notation of representation errors from $ERR_{o\_r}$ to $ERR_{REP}$ and included this term in Figure 1. as follow:

"First, we define two quantities: 1) the root mean square errors (*RMSE*) between the ensemble mean of posterior $CO_2$ concentrations and observed $CO_2$ concentrations, and 2) $ERR_{TOT}$ (Section 2.3). $RMSE^2$ represents the true errors in OCO-2 MIP ensemble mean of $CO_2$ concentrations including representation errors ($\sigma_r^2$), observation errors ($\sigma_o^2$), true flux errors projected onto $CO_2$ concentration ($\sigma_{f_t}^2$), transport errors ($\sigma_t^2$), and error covariances between the preceding two terms ($cov(\sigma_{f_t}, \sigma_t)$). $ERR_{TOT}^2$ is the sum of the estimated error components, defined as the sum of $ERR_{REP}^2$, $ERR_{OBS}^2$ and $ERR_{MIP}^2$. $ERR_{REP}^2$ and $ERR_{OBS}^2$ indicate representation errors ($\sigma_r^2$) and observation errors ($\sigma_o^2$), respectively. $ERR_{MIP}^2$ is the sum of estimated flux errors projected onto $CO_2$ space ($\sigma_{f_e}^2$) and transport errors ($\sigma_t^2$), and their error covariances ($cov(\sigma_{f_e}, \sigma_t)$), computed from an ensemble spread of posterior $CO_2$ concentrations. Here we separate representation errors from transport errors for computational purpose. The ratio between $ERR_{TOT}$ and $RMSE$ is then used to evaluate whether the estimated flux errors, computed from the ensemble spread of posterior fluxes, overestimate or underestimate the true errors in the ensemble mean fluxes. Next, we calculate the estimated flux errors projected onto atmospheric $CO_2$ ($h(err_{f_e})$) through atmospheric transport simulations (Section 2.4). With $h(err_{f_e})$, $ERR_{TOT}$, and $RMSE$, we derive the true errors in ensemble mean of posterior fluxes projected onto $CO_2$ space ($h(err_{f_t})$). Then, we identify the areas where these airborne observations are most sensitive to using an adjoint sensitivity analysis and calculate the estimated posterior flux errors over these regions ($err_{f_e}$). Assuming a linear observation operator, the study finally computes the true errors of the ensemble mean posterior fluxes over the identified sensitive areas ($err_{f_t}$) by applying the ratio between $h(err_{f_t})$ and $h(err_{f_e})$ to $err_{f_e}$."

[Figure]

*Figure 1: Flow chart summarizing the process of evaluating and quantifying errors in ensemble mean of regional posterior fluxes. $RMSE^2$ is the mean square errors between the ensemble mean of posterior $CO_2$ concentrations and observed $CO_2$ concentrations. $ERR_{REP}^2$ and $ERR_{OBS}^2$ denote estimates of observation errors and representation errors, respectively. $ERR_{MIP}^2$ is an ensemble spread of posterior $CO_2$ concentrations. $ERR_{TOT}^2$ is defined as the sum of $ERR_{REP}^2$, $ERR_{OBS}^2$, and $ERR_{MIP}^2$. $err_{fe}$ and $err_{ft}$ are estimates of flux errors, defined as an ensemble spread of posterior fluxes, and their true values. $h(err_{fe})$ and $h(err_{ft})$ are estimates of flux errors projected onto $CO_2$ concentrations and their true values. $\sigma_o^2$, $\sigma_r^2$, $\sigma_{ft}^2$ $(\sigma_{fe}^2)$, $\sigma_t^2$, and $cov(\sigma_{ft}, \sigma_t)$ indicate the types of errors represented by the error statics, namely observation errors, representation errors, true (estimated) flux errors projected onto $CO_2$ concentration, transport errors, and error covariances between the preceding two terms, respectively.*

11. Line 126.  I would recommend adding citations that document these field campaigns.

 [Response] We include citation information for the measurement campaigns in both the text and Table 1 of the revised manuscript.

"The dataset includes two airborne measurement campaigns (Atmospheric Tomography Mission; ATom; **Thompson et al. 2022** and $O_2/N_2$ Ratio and $CO_2$ Airborne Southern Ocean Study; ORCAS; **Stephens et al. 2018**) over the ocean, as well as 18 campaigns over land."

12. Figure 2. The caption refers to the number of airborne measurements.  What constitutes one airborne measurement?  Many aircraft campaigns have continuous observations and gigabytes of data.  Please explain the quantization of the data that is used in this figure.  If this number of the number of 1x1 degree grids with an

observation, what is the temporal unit for an observation?  If the same location is measured for 100 hours over 10 days within one month, is that one measurement or ten or 100 measurements?

[Response] Figure 2a illustrates the total number of airborne measurements data used in our analysis. Figure 2b shows the number of 1°x1° grid-points where more than 10 observations were made within each region for every month. If observation is made at the same location (i.e., same grid point) 100 hours over 10 days within one month, it is considered as one grid point. We revise the caption of Figure 2 to convey their definition more clearly:

*"Figure 2: (a) Total number of airborne measurement data used in this study at each 1°×1° grid point and (b) the number of 1°×1° grid-points, where more than 10 data is available, within each region and each month for the period 2015–2017."*

13. Table 1.  Please include citations for data sets when possible.  I am sure, for example, that there is a data citation available for AToM observations.

[Response] We include citation information for all data sets in Table 1 of the revised manuscript.

14. Line 149.  "simulated atmospheric CO2 mole fractions"?  And please explain, "the observed one".  What is "the observed one"?

[Response] We revised these sentences to convey our approach more clearly as follow:
"One is RMSE between the ensemble mean of posterior atmospheric $CO_2$ from OCO-2 MIP models and the atmospheric $CO_2$ from airborne measurements, …"

15. Line 153. "the 1x1 grid cell"

[Response] We revised it as follow:
"… within **each** 1°×1° grid-cell in each month …"

16. Line 153.  What constitutes one airborne measurement?  The continuous aircraft campaigns have MANY more measurements than is suggested by Figure 2.  Please explain your definition of one measurement.

[Response] In this study, all error statistics such as RMSE are computed using airborne measurement data made within each 1°x1° grid-cell during a month. In other words, all airborne measurement data recorded within a single grid point for one month provide one measurement information for evaluating inversion estimates. We revised that sentence in the revised manuscript:

"$\overline{h_i(\hat{x})}$ is the ensemble mean of posterior atmospheric $CO_2$ sampled at the time and location of the $i$th airborne observation $y_{o,i}$, within each 1°×1° grid-cell in each month. $N$ is the monthly total number of sampled data at each grid-cell. $M$ is the number of ensemble members (i.e., 10). A single monthly *RMSE* value is computed using $N$ measurement data at each grid-cell. The number of *RMSE* values is calculated per month within each region corresponds to the number of grid-cells shown in Figure 2b."

17. Line 159-160.  I don't believe that the ensemble mean accounts for transport errors.  The ensemble includes them, at least as represented by the ensemble.

[Response] We agree with the reviewer's comment. Transport errors can be estimated from the ensemble spread of inverse model estimates, rather than from the ensemble mean values. We revised the sentence and relocated it to the part where $ERR_{MIP}^2$ (ensemble spread of posterior CO2 concentrations) is introduced (line 194–197 in the revised manuscript).

"Different from Liu et al. (2021) which used only one transport model, $ERR_{MIP}^2$ accounts transport errors because posterior atmospheric $CO_2$ were generated by multiple types of transport models in OCO-2 MIP driven by different meteorology fields. Thus, $ERR_{MIP}^2$ term accounts for transport errors, but not representation errors due to the coarse spatial resolution of these transport models with the highest spatial resolution being 2°×2.5°."

18. Line 161.  I don't object to removing these outliers, but I'm not sure this ensures robust error estimates.

[Response] We eliminated this process which excludes outliers and re-calculated all error quantities by using all observation data in the analysis. Since outliers comprised 0.05% of the total data, the newly computed results, particularly the ratio of three-year mean ERR$_{TOT}$ and RMSE and the ratio of three-year mean $h(err_{f_e})$ to $h(err_{f_t})$, key metrics for assessing and quantifying regional terrestrial biosphere flux errors, do not exhibit significant differences compared to the previous results (Table R1).

*Table R2 Mean values of the regionally averaged ratios of ERR$_{TOT}$ to RMSE and the ratios of $h(err_{f_e})$ to $h(err_{f_t})$ for 2015–2017 with their 95% confidence intervals derived from 1000 bootstrap samples of datasets, calculated using atmospheric CO$_2$ datasets within the range of 1-5 km above ground level when excluding (previous results) or including outliers (revised results).*

| | | Alaska | Mid NA | Europe | East Asia | Southeast Asia | Australia | South America |
|---|---|---|---|---|---|---|---|---|
| Previous results | ERR$_{TOT}$/RMSE | 0.98 [0.89, 1.08] | 0.91 [0.84, 0.97] | 0.79 [0.61, 0.97] | 0.87 [0.81, 0.94] | 0.75 [0.65, 0.86] | 0.73 [0.59, 0.87] | 1.03 [0.83, 1.28] |
| | $h(err_{f_e})/h(err_{f_t})$ | 0.96 [0.76, 1.17] | 0.75 [0.61, 0.90] | 0.52 [0.28, 0.78] | 0.64 [0.53, 0.77] | 0.56 [0.41, 0.72] | 0.59 [0.34, 0.87] | 1.10 [0.51, 1.79] |
| Revised results | ERR$_{TOT}$/RMSE | **0.98 [0.89, 1.08]** | **0.90 [0.83, 0.97]** | **0.79 [0.61, 0.97]** | **0.84 [0.78, 0.91]** | **0.75 [0.65, 0.86]** | **0.73 [0.59, 0.87]** | **0.99 [0.79, 1.24]** |
| | $h(err_{f_e})/h(err_{f_t})$ | **0.96 [0.76, 1.17]** | **0.74 [0.61, 0.88]** | **0.52 [0.27, 0.78]** | **0.59 [0.48, 0.70]** | **0.56 [0.41, 0.72]** | **0.59 [0.34, 0.87]** | **0.97 [0.49, 1.54]** |

19. Line 175-180.  I may just be tired, but I am having a very hard time following this discussion.  This is an interesting approach to evaluating uncertainty.  It would be great if this could be explained more clearly.  Figure 1 is an interesting complement to this text, but it isn't cited at all in this text.  Perhaps you could clarify your methods by connecting the terms in Figure 1 explicitly to this text and to Appendix A.

[Response] We appreciate the reviewer's suggestion. In the revised manuscript, we changed the notation of representation errors from ERR$_{o\_r}$ to ERR$_{REP}$ and included this term in Figure 1. We also revised sentences to clearly convey our approach for estimating representation errors:

"To obtain representation errors and observation errors not captured by $ERR_{MIP}^2$, we additionally calculate $ERR_{REP}^2$ and $ERR_{obs}^2$, respectively. $ERR_{REP}^2$ indicates the representation errors ($\sigma_r^2$) in $RMSE^2$ as shown in Fig. 1 and is defined as a spatial variability of atmospheric $CO_2$ within a 2°×2.5° grid cell written as:

$$ERR_{REP}^2 = \frac{1}{N}\sum_{i=1}^{N} VAR_{CO_2,i} \tag{3}$$

With the high-resolution (0.5º×0.625º) 3-hourly GEOS-5 simulation results for 2018 from NASA Goddard Space Flight Center (Weir et al., 2021), we calculate the variance of atmospheric $CO_2$ concentration within each 2°×2.5° grid cell at every 3-hour interval. Then, we sample the $CO_2$ variance value ($VAR_{CO_2,i}$) at the grid cell containing the i$^{th}$ observation and the time closest to the observation. Subsequently, the monthly mean values of the $N$ co-sampled variances are derived ($ERR_{REP}^2$)."

20. Line 202. Please explain, "the regional average of error matrices."

[Response] we revised that sentence as follow:

"By applying 1000 bootstrap resampling to the monthly grid-based error statistics (e.g., $RMSE$, $ERR_{MIP}$, $ERR_{REP}$, and $ERR_{TOT}$) within each region, we obtain regional mean values of these error statistics, along with their corresponding 95% confidence intervals."

21. Figure 2a. Some regions have very, very few observations. What does that do to your results?

[Response] The availability of observation data in each region impacts both the reliability of our error statistics for quantifying regional flux errors and the area extents represented by these statistics. The monthly true flux error ($h(err_{f_t})$) is calculated using the Eq. 9, $h(err_{f_t})^2 - h(err_{f_e})^2 = RMSE^2 - ERR_{TOT}^2$. Out of 181 cases, representing the total months of observation across all seven regions, $h(err_{f_t})$ can be derived using this equation in 158 cases. However, in 23 cases (13% of total cases), $h(err_{f_t})$ cannot be derived from this calculation method when ERR$_{TOT}$ and/or $h(err_{f_e})$ values fell outside the applicable range (Figure 5a-g). Around 40% of the exception cases occur in South America where observation data cover 1-6 grid cells by month. This indicates that observation data are insufficient to quantify the monthly flux errors in this region. In addition, the limited data availability results in a larger uncertainty range of the ratios of three-year mean $h(err_{f_e})$ to $h(err_{f_t})$, which are key metrics for quantifying regional terrestrial biosphere flux errors (Figure 5h). For example, the uncertainty ranges of the 95% confidence interval are 0.51, 0.53 and 1.05 for $h(err_{f_e})$ to $h(err_{f_t})$ ratios in Europe, Australia, and South America respectively, while the uncertainty ranges are 0.28 and 0.22 in mid-latitude north America and East Asia respectively, where observations cover wider areas and occur more frequently. Lastly, to identify areas that primarily contribute to the computed three-year mean error statistics, we considered regions that were selected as effective areas for at least eight months or more (Figure 6a; outlined area). Our error statistics computed from data in mid-latitude North America and East Asia represent broad regions encompassing the United States and eastern parts of East Asia. In contrast, those computed from data in Alaska and Europe, where observation made for limited periods and at specific locations, represent much smaller areas.

These results highlight the importance of frequent airborne measurements with extensive spatial coverage for reliable quantification of errors in regional terrestrial flux estimates derived from inverse models. We included above explanations in the revised manuscript (lines 275-279, 448-450, and 513-527).

22. Lines 312-313.  Are the RMSE values between 1 and 3 ppm?  Or is 1-3ppm the range of the values of RMSE?

[Response] We revised that sentence as follow:
"RMSE values in all these regions exhibit significant monthly variations, **with values falling within the range of 1-3 ppm,** with no clear seasonality possibly due to variations in observation routes (Figure 4)."

23. Figure 4. caption.  I think these are monthly values of RMSE.  Monthly variations of RMSE sounds to me like the variance of the RMSE.

[Response] we revised that expression in Figure 4 and Figure 5 as follow:

"Figure 4: (a-g) **Monthly values** of RMSE, …"
"Figure 5: (a-g) **Monthly values** of $h\left(err_{f_e}\right)$ and …"

24. Paragraph starting on line 336. What is the main point of this paragraph?  I have the same concern for all the paragraphs up to line 384.  These paragraphs tend to describe the contents of the figures.  It is hard for me to extract the main result.  I suggest starting each of these paragraphs with a topic sentence that presents your main finding, then use the paragraph to explain this finding.

[Response] We appreciate the reviewers' comments. We revised these paragraphs shown in line 374–415 in the revised manuscript.

25. Line 386.  I cannot find in section 2 where to find the method for determining the most influential areas for observed atmospheric CO2. And again, this is not a result.\

26. Line 393-394.  I do not understand what is meant by the sentence starting with "Figure 6a…" and I don't understand the associated figure.  Further, the text following this statement describes methodology, not results.  Can you please explain Figure 6 methodology in the methods section of the text?

[Response] We agree with the reviewer's comments regarding the paragraph containing sentences (Line 386, Line 393-394), which described the methodology. We have relocated this content to the methodology section and revised the sentences to better convey our approach. The methods for determining the most influential areas for observed atmospheric CO2 and for deriving Figure 6 are now described in lines 297–322 in the revised manuscript.

27. Lines 419-421.  I don't understand how this follows from the preceding text.  If this is the main finding, please begin the paragraph with this statement, then use the paragraph to explain this statement.  At present, I cannot follow this argument.  It is an interesting argument.  Please explain it more clearly.

[Response] We revised the paragraph by reorganizing the sentences and incorporating information on anthropogenic emissions as follows:

"Finally, by using the three-year regional mean ratios between $h\left(err_{f_e}\right)$ and $h\left(err_{f_t}\right)$, we compute the true errors in the annual terrestrial fluxes over the effective areas averaged for the period 2015–2017 (Figure 6). We find that the actual terrestrial biosphere flux errors are underestimated, particularly in regions where annual $CO_2$ emissions from fossil fuel combustion exceed annual terrestrial biosphere fluxes by 3-31 times. The airborne measurements carried out in mid-latitude North America, East Asia, and Southeast Asia are influenced by a broad region encompassing the United States, the eastern part of East Asia, and the western part of Southeast Asia where fossil fuel $CO_2$ emissions are 1,341, 2,443, and 815 Tg C year$^{-1}$, respectively. The first two regions are estimated as significant terrestrial biosphere $CO_2$ sinks, with estimated fluxes of −414 ± 279 (ensemble mean ± 1σ) and −561 ± 380 Tg C year$^{-1}$, in contrast to Southeast Asia (26 ± 118 Tg C year$^{-1}$). However, the $CO_2$ sinks are more than 3 and 4 times smaller than the fossil fuel $CO_2$ emissions, respectively. The recalculated terrestrial biosphere flux errors in these regions exceed the ensemble spread with values of 374, 643, and 211 Tg C year$^{-1}$. Observations in Europe and Australia, conducted over limited periods and specific locations, mainly represent certain areas in the western Europe and the southeastern part of Australia, where fossil fuel emissions (234 and 53 Tg C year$^{-1}$, respectively) are around four and five times greater than terrestrial biosphere sinks (−51 ± 34 and −10 ± 67 Tg C year$^{-1}$). The recalculated terrestrial biosphere flux errors in these regions are also larger than the ensemble spread, estimated at 65 and 114 Tg C year$^{-1}$, respectively. On the contrary, the most influential areas for the observation in Alaska and South America, encompassing the southeastern region of Alaska and the northern part of Brazil, characterized as a terrestrial biosphere sinks of −8 ± 11 Tg C year$^{-1}$ and sources of 625 ± 387 Tg C year$^{-1}$, respectively, which are comparable to or more than 10 times greater than fossil fuel emissions (10 and 38 Tg C year$^{-1}$). The observation-based estimates of true terrestrial biosphere flux errors are almost identical to the ensemble spread in both regions with values of 11 and 398 Tg C year$^{-1}$, respectively."

[Figure]

**Figure 6:** (a) Number of months selected as the effective area for airborne measurements. The outlined area represents selected areas for more than eight months or equal. (b) Annual total terrestrial biosphere $CO_2$ flux obtained from the ensemble mean of ten OCO-2 MIP models and annual total fossil fuel $CO_2$ emissions estimated from ODIAC data for each outlined area averaged over the period 2015–2017. The error bars in black and red indicate the one standard deviation of the inversion estimates and the newly estimated error range from this study, respectively.

28. Lines 424-428. This is material for the introduction, not the discussion.

[Response] We delete these sentences in the revised manuscript.

29. Line 435-437.  This sentence needs work.

[Response] We revised that sentence as follow:
"For example, although the three-year mean errors in representation and transport in East Asia exceed those in Southeast Asia by 0.5 and 0.3 ppm, the disparity in projected mean true flux errors onto $CO_2$ space between the two regions is only 0.2 ppm."

30. Line 437. This result could be a natural consequence of what?

[Response] We revised that sentence as follow:

"This result is supported by previous studies highlighting that the spatial distributions of simulated $CO_2$ concentrations can vary significantly depending on the transport model (Schuch et al., 2023) and their spatial resolution (Stanevich et al., 2020)."

31. Line 444.  I don't follow the "This underestimation…" sentence.  Please clarify.

32. Line 444. What do you mean by "the common assumptions and observations…"?  Are you arguing that since many ensemble members share common data and common methodological assumptions, this results in the spread among them being an underestimate of the true uncertainty in fluxes?  This is possible and an interesting assertion, but I don't think it is proven by this work.

[Response] We agree with the reviewer' comment. This study does not specify major reasons for the underestimation in true flux errors, but discusses possible causes for the underestimation. We revised that sentence to clearly convey our intention as follow:

"The underestimation of true flux errors can arise from multiple factors, posing a challenge in determining main cause of the underestimation. Possible reasons include errors in methodological assumptions and atmospheric $CO_2$ observation data commonly applied to all OCO-2 MIP ensemble members because flux errors arising from these components are not captured by the ensemble spread."

33. Line 449.  What is a "main source region"?

[Response] We revised it as follow:

"The underestimation of true flux errors only in regions **with more than three times greater fossil fuel emissions than biosphere fluxes** suggests …"

34. Lines 458-459.  I don't understand the origins of the 15% figure, or the meaning of "challenges" in estimating monthly  flux errors.  I very much agree with the concern at the end of this paragraph that areas with limited data

may not have sufficient data for computing reliable error statistics for the flux inversions. I think these are related topics. Please clarify.

[Response] Right. This paragraph discusses the impact of regionally different observation data availability on our results. We revise that paragraph as follow:

"The reliability of our observation-based regional flux error estimates is based upon the data availability of airborne measurements. Although our approach is generally effective in estimating a regional mean of monthly $h(err_{f_t})$, it is not applicable in 15% of our total cases (shown in Figure 5), when measurements were mostly made in local areas covering one to six 1°×1° grid cells within each region. This limitation may be attributed to the application of a common method for calculating observation errors across all data points, which might not adequately identify specific outliers. Caution is required when applying our approach to monthly-scale analysis, especially when using observations made locally. Extending the calculation period to several months or longer (e.g., Figure 5h) is a suitable strategy for mitigating the impact of outliers and obtaining more robust results. In fact, the ratios of three-year mean $h(err_{f_e})$ to $h(err_{f_t})$, which are key metrics for quantifying regional flux errors (Figure 5h), have a smaller uncertainty in mid-latitude North America and East Asia where wide and consistent airborne data are available, than over Europe and South America, where aircraft observations are sparse and only have intermittent data coverage. In addition, it is noteworthy that the $h(err_{f_e})$ to $h(err_{f_t})$ ratios derived from continuous observations enable the computation of unbiased true errors in the average annual ensemble terrestrial fluxes for the analysis period, compared to those from limited observation periods (e.g., in Alaska). These results highlight the importance to have frequent airborne measurements with extensive spatial coverage for the reliable error quantification of regional terrestrial flux estimates derived from inverse models."

35. Line 477. "Second…" This is another paragraph.

[Response] We separated the paragraph in the revised manuscript.

36. Line 491-492. I am not convinced that these flux errors are largest in errors with large anthropogenic fluxes. This is a plausible hypothesis, but I would not say that the results reveal this to be true. I would like to see a more careful analysis of the fossil fluxes in the relevant influence regions, and the relationship between the strength of fossil fluxes and these flux errors to be convinced.

[Response] We agree with the reviewer's comment. We add information on fossil fuel emissions to support our main finding, "the actual errors in ensemble mean of annual terrestrial biosphere flux estimates of OCO-2 MIP are underestimated, particularly in regions with higher fossil fuel CO2 emissions compared to terrestrial biosphere CO2 fluxes" in the result part in the revised manuscript (lines 452–471; or please refer to our response to your 27th comment).

---

## Author Response (AR2)

The authors revised their text extensively, but three points have not been properly addressed, the first two being major. The line numbers below are those used in my initial review.

[Response] We appreciate your constructive comments on this manuscript. We revised the manuscript to fully address your comments and suggestions. Detailed point-by-point responses to your comments and related revisions are presented below. The original comments are in black, and our responses are in blue color.

L27 - I understand that the authors do not "conclude that it [one element of the MIP protocol about fossil fuel emissions] is the main source of the underestimation in the true flux errors". But this element in the protocol is a likely explanation that, if correct, would make "the main finding of the study" (authors' words) pretty useless.

[Response] During the previous round of review, we were unsure about the meaning of this comment, "this main conclusion is not a finding but an input to the study". Upon further reflection, we believe that the reviewer is pointing out that the errors we estimated are on the net surface-atmosphere $CO_2$ fluxes, which is the combined terrestrial biosphere $CO_2$ fluxes and fossil $CO_2$ emissions. This is an insightful criticism of the way our results were presented. We have re-oriented the manuscript to emphasize the error estimates are on the net $CO_2$ fluxes, not just the terrestrial biosphere $CO_2$ fluxes. We want to emphasize that this reinterpretation does not impact our results, as the ensemble spread of posterior terrestrial biosphere fluxes is identical to that of posterior net fluxes. This is because all OCO-2 MIP models prescribed the same fossil fuel emission estimates and treated them perfectly known values. We revised the manuscript throughout, changing the previously stated "errors in the terrestrial biosphere $CO_2$ flux estimates" to "errors in the net $CO_2$ flux estimates" and sentences related to these revisions. The sentences with major revisions related to this are as follows:

L69-71, "This study aims to develop a framework to quantify the errors in **regional net surface-atmosphere $CO_2$ fluxes (terrestrial biosphere fluxes + fossil fuel emissions)** estimated from an ensemble of inverse models by using airborne $CO_2$ measurements, transport modeling, and adjoint sensitivity analysis."

L313-318, "Therefore, we can obtain the true errors in the ensemble annual total **net land fluxes** in those areas, $err_{f_t}$ ($= \sigma_{f_t}$), by multiplying the ratio between three-year mean values of $h\left(err_{f_t}\right)$ and $h\left(err_{f_e}\right)$ by the ensemble spread of the **annual total net land flux estimates** ($err_{f_e}$) within the effective areas. The equation can be written as:

$$err_{f_t} = \frac{h\left(err_{f_t}\right)}{h\left(err_{f_e}\right)} \times err_{f_e} \tag{12}$$

**One thing readers should keep in mind is that the $err_{f_e}$ is identical to the ensemble spread of posterior terrestrial biosphere fluxes because all OCO-2 MIP models used uniform fossil fuel emission estimates and assumed them to be perfectly known.**"

L467-469, "The black error bars denote ± one standard deviation of the **posterior net land fluxes, identical to those of the posterior terrestrial biosphere fluxes.** The error bars in red indicate the newly-estimated range of errors **in the posterior net land fluxes** from this study."

In addition, we removed the first paragraph of the original introduction, which emphasized terrestrial biosphere flux estimates. The **title** of this study is also revised as follows:
"Quantification of regional **net** $CO_2$ flux errors in the v10 OCO-2 MIP ensemble using airborne measurements"

Moreover, we clearly described the reason for the underestimation of the true flux errors in the regions with high fossil fuel emissions and the implications of our study in the main text. The revised sentences are as follows:

L25-26, "This suggests the presence of systematic biases in the inversion estimates associated with errors in the prescribed fossil fuel emissions common to all models."

L487-501, "Our analysis reveals that the true errors in the ensemble mean of posterior net $CO_2$ flux estimates is significantly greater than the ensemble spread of flux estimates in five out of seven regions with higher fossil fuel emissions compared to terrestrial biosphere fluxes. Possible explanation for this result is the presence of errors in the prescribed fossil fuel emissions common to all OCO-2 MIP models. OCO-2 MIP models treated fossil fuel emissions as perfectly known values and adjusted terrestrial biosphere and ocean $CO_2$ fluxes to minimize the difference between the simulated and observed $CO_2$ concentrations. Thus, if there are errors in the prescribed fossil fuel emission estimates, these errors propagate into the posterior natural flux estimates. The assumption used in the OCO-2 MIP models is, in fact, the one often applied in conventional global atmospheric inverse models as it is considered that the errors in fossil fuel emission estimates are relatively lower than those in natural flux estimates at national scales (4-20%; Andres et al., 2014). However, the emission errors become substantial when considering spatial distribution at model grid scale and temporal variability within a year (Zhang et al., 2016; Gurney et al., 2021). Oda et al. (2023) showed significant impacts of differences in fossil fuel emission estimates on posterior terrestrial biosphere flux estimates near the source regions. OCO-2 MIP models used identical fossil fuel emission estimates and thus their posterior net flux estimates share common biases induced by the errors in the fossil fuel emission estimates. Because these systematic biases are not captured by the ensemble spread of flux estimates, true flux errors exceed the errors computed from the ensemble spread in the main source regions. In addition to this, …"

L604-608, "… This result provides observation-based evidence supporting previous studies (Oda et al., 2023; Wang et al., 2020) that emphasized the impact of fossil fuel emission errors on global atmospheric $CO_2$ inversions. This finding offers important insights into understanding the sources of errors in current inverse modeling and highlights the need for improving fossil fuel emission estimates and developing inversion methods that account for uncertainties in both fossil fuel emissions and natural fluxes. …"

− Andres, R. J., Boden, T. A., and Higdon, D.: A new evaluation of the uncertainty associated with CDIAC estimates of fossil fuel carbon dioxide emission, Tellus B: Chem. Phys. Meteorol., 66, 23616, https://doi.org/10.3402/tellusb.v66.23616, 2014.
− Gurney, K. R., Liang, J., Roest, G., Song, Y., Mueller, K., and Lauvaux, T.: Under-reporting of greenhouse gas emissions in U.S. cities, *Nat. Commun.*, 12, 553, https://doi.org/10.1038/s41467-020-20871-0, 2021.
− Oda, T., Feng, L., Palmer, P. I., Baker, D. F., and Ott, L. E.: Assumptions about prior fossil fuel inventories impact our ability to estimate posterior net $CO_2$ fluxes that are needed for verifying national inventories. *Environ. Res. Lett.*, *18*(12), 124030, https://doi.org/10.1088/1748-9326/ad059b, 2023.
− Wang, J. S., Oda, T., Kawa, S. R., Strode, S. A., Baker, D. F., Ott, L. E., and Pawson, S.: The impacts of fossil fuel emission uncertainties and accounting for 3-D chemical $CO_2$ production on inverse natural carbon flux estimates from satellite and in situ data, *Environ. Res. Lett.*, *15*(8), 085002, https://doi.org/10.1088/1748-9326/ab9795, 2020.
− Zhang, X., Gurney, K. R., Rayner, P., Baker, D., and Liu, Y.-P.: Sensitivity of simulated $CO_2$ concentration to sub-annual variations in fossil fuel $CO_2$ emissions, *Atmos. Chem. Phys.*, 16, 1907–1918, https://doi.org/10.5194/acp-16-1907-2016, 2016.

L220 – I respectfully disagree because sub-monthly flux patterns may also affect monthly concentration patterns, depending on sub-monthly transport patterns. This point needs to be documented.

[Response]We agree the reviewer's comment. We included following sentences to carefully discuss the possible impact of variability of sub-monthly flux variations within OCO-2 MIP models on our results in the main text (L587-599):

"This study uses monthly mean posterior flux estimates for the calculation of monthly $h\left(err_{f_e}\right)$ but posterior flux estimates from each OCO-2 MIP models have different sub-monthly patterns. This could modify the sub-monthly variations in posterior atmospheric $CO_2$ and affect the ensemble spread of posterior $CO_2$ concentrations. However, due to absence of information on sub-monthly variations in posterior flux estimates, this study assumed that the contributions of the inter-model variability of sub-monthly flux variations to our monthly mean error quantities ($h\left(err_{f_e}\right)$ and $ERR_{MIP}$) are not significant. This assumption is supported by comparing $h\left(err_{f_e}\right)$ with $ERR_{MIP}$. $ERR_{MIP}$ resulted from variabilities in not only hourly posterior flux estimates but also transport models. Despite $h\left(err_{f_e}\right)$ not accounting for the impacts of inter-model variability of sub-monthly flux patterns, the regional mean of monthly $h\left(err_{f_e}\right)$ (0.44-0.93 ppm), on average, accounts for 58-86% of regional mean of monthly $ERR_{MIP}$ (0.51-1.34 ppm) throughout the analysis period (Figure 5h and Fig. S6). Furthermore, we found that our main results remain robust across the potential range of $h\left(err_{f_e}\right)$ when it includes the impact of sub-monthly flux variations. For example, if $h\left(err_{f_e}\right)$ increases, on average, by 0.2 ppm, the ratio of regional mean of $h\left(err_{f_e}\right)$ to $h\left(err_{f_t}\right)$ increases from 0.74 [0.61, 0.88] to 0.83 [0.71, 0.96] in midlatitude North America and from 0.59 [0.48, 0.70] to 0.67 [0.57, 0.78] in East Asia throughout the analysis period."

L300 - CMS-Flux is not "most models". The last part of the sentence ("suggesting…") should be deleted.

[Response] We agree that the previous reference does not represent "most models". Instead, we cited Gaubert et al. (2023), which revealed that most OCO-2 MIP inverse models overestimate the observed atmospheric $CO_2$ concentrations along the African coast during the ATom project, due to potential biases in OCO-2 $XCO_2$ measurements over northern tropical Africa. We also revised the corresponding sentence accordingly (L362-363) as follows:

"These findings agree with Gaubert et al. (2023), which showing most of the inverse models in v10 OCO-2 MIP have significant errors because of potential positive biases in OCO-2 $XCO_2$ measurements for this region."

– Gaubert, B., Stephens, B. B., Baker, D. F., Basu, S., Bertolacci, M., Bowman, K. W., Buchholz, R., Chatterjee, A., Chevallier, F., Commane, R., Cressie, N., Deng, F., Jacobs, N., Johnson, M. S., Maksyutov, S. S., McKain, K., Liu, J., Liu, Z., Morgan, E., O'Dell, C., Philip, S., Ray, E., Schimel, D., Schuh, A., Taylor, T. E., Weir, B., van Wees, D., Wofsy, S. C., Zammit-Mangion, A., and Zeng, N.: Neutral Tropical African CO2 Exchange Estimated From Aircraft and Satellite Observations, Global Biogeochem. Cycles, 37, e2023GB007804, https://doi.org/10.1029/2023GB007804, 2023.

In addition, the title now needs "the" before "v10 OCO-2 MIP ensemble".

[Response] Thank you. We revised the title as follows:

Quantification of regional net $CO_2$ flux errors in **the** v10 OCO-2 MIP ensemble using airborne measurements

---

## Author Response (AR3)

[Response] We appreciate your constructive comments on this manuscript. We revised the manuscript to fully address your comments and suggestions. Detailed point-by-point responses to your comments and related revisions are presented below. The original comments are in black, and our responses are in blue color.

The authors have worked on my remarks seriously again, but the answer to point "L220 – I respectfully disagree because sub-monthly flux patterns may also affect monthly concentration patterns, depending on sub-monthly transport patterns. This point needs to be documented" is weak. I regret to have to recommend further analysis of this potential weakness of the study.
On a minor side "due to absence of information on sub-monthly variations in posterior flux estimates" is not correct (this information is available somewhere, even though it may not be public).
[Response] We agree with the reviewer's comment. We conducted an additional analysis using various datasets of hourly (or 3-hourly) terrestrial biosphere fluxes to assess how different sub-monthly patterns of posterior flux estimates from OCO-2 MIP models might affect our main results. The revisions based on this additional analysis are as follows (L587-608 in the revised manuscript):

"This study uses monthly posterior flux estimates to calculate monthly $h\left(err_{f_e}\right)$. However, posterior flux estimates from each OCO-2 MIP model have different sub-monthly patterns, which could modify sub-monthly variations in posterior atmospheric $CO_2$ concentrations and, in turn, affect their ensemble spread. To examine their potential impact on the results, we conduct an analysis with different publicly available hourly (or 3-hourly) terrestrial biosphere fluxes (Chevallier et al., 2019; Jacobson et al., 2020; Ott et al., 2020; Haynes et al., 2021; Liu and Bowman, 2024) which are from seven OCO-2 MIP prior flux models (Ames, Baker, CAMS, CMS-Flux, CT, OU, and WOMBAT; Table S1). By incorporating the monthly-balanced hourly flux estimates into the monthly posterior fluxes, we generate hourly posterior terrestrial biosphere flux estimates for these seven models. Since the assimilation window for each OCO-2 MIP model ranges from one week to one month, the weekly variations in posterior fluxes may differ from those in the prior fluxes. Nonetheless, with only the monthly posterior flux estimates being publicly available, this approach offers valuable insights into how different sub-monthly patterns of posterior fluxes could affect our main results. Our analysis shows that the regional averages of $h\left(err_{f_e}\right)$ derived from the monthly posterior flux estimates from the seven models are, on average, within ±10% of the values originally obtained using flux estimates from 10 models for the period 2015–2017, except for Europe (13% lower) (Fig. S12a). When accounting for different sub-monthly patterns of posterior fluxes across models, the regional averages of $h\left(err_{f_e}\right)$ increase by 10-22% (0.06-0.14 ppm) across six regions, with a 45% (0.23 ppm) increase in Europe. These results suggest that our earlier calculation, assuming identical sub-monthly flux variations, underestimates $h\left(err_{f_e}\right)$. We further investigate whether our main finding remains robust even if we adjust the original values of $h\left(err_{f_e}\right)$ using the potential underestimation rate. After making the correction, we found that the ratios of the regional average $h\left(err_{f_e}\right)$ to $h\left(err_{f_t}\right)$ increase the most in Europe by 0.14 and only up to 0.07 in the other six regions, as $h\left(err_{f_t}\right)$ also rises with $h\left(err_{f_e}\right)$ according to Eq. (8) (Fig. S12b). Moreover, the $h\left(err_{f_t}\right)$ still exhibits significant underestimation (p<0.05) in mid-latitude North America, Europe, East Asia, Southeast Asia, and Australia. This indicates that our main results are robust to the inclusion or exclusion of sub-monthly flux patterns in the calculation of $h\left(err_{f_e}\right)$."

[Figure]

**Figure S12. (a)** Mean values of monthly $h(err_{f_e})$ for each region for the period 2015–2017, derived using different monthly posterior flux estimates from either 10 (blue) or seven ($(h(err_{f_e}))_7$; light gray) OCO-2 MIP models with identical hourly NBE variation information, or from different monthly posterior flux estimates from seven models with different hourly NBE variation information ($(h(err_{f_e}))_{7\_hourly}$; dark gray). The numbers at the top of the panel (a) indicate the ratio of $(h(err_{f_e}))_{7\_hourly}$ to $(h(err_{f_e}))_7$. **(b)** Mean monthly values of $h(err_{f_e})$ (blue) and $h(err_{f_t})$ (red) for each region over three years, with the dotted bars representing the corrected $h(err_{f_e})$, obtained by multiplying the ratio of $(h(err_{f_e}))_{7\_hourly}$ to $(h(err_{f_e}))_7$, and the recalculated $h(err_{f_t})$ using these corrected $h(err_{f_e})$ values. The numbers at the top of the panel (b) denote the ratio of $h(err_{f_e})$ to $h(err_{f_t})$. The error bars represent the 95% confidence intervals derived from 1000 bootstrap samples of datasets.

- Chevallier, F., Remaud, M., O'Dell, C. W., Baker, D., Peylin, P., and Cozic, A.: Objective evaluation of surface- and satellite-driven carbon dioxide atmospheric inversions. *Atmospheric Chemistry and Physics*, *19*(22), 14233-14251, https://doi.org/10.5194/acp-19-14233-2019, 2019.
- Haynes, K.D., I.T. Baker, and A.S. Denning.: SiB4 Modeled Global 0.5-Degree Hourly Carbon Fluxes and Productivity, 2000-2018, ORNL DAAC, Oak Ridge, Tennessee, USA, https://doi.org/10.3334/ORNLDAAC/1847, 2021.
- Jacobson, A. R., Schuldt, K. N., Miller, J. B., Oda, T., Tans, P., Arlyn Andrews, Mund, J., Ott, L., Collatz, G. J., Aalto, T., Afshar, S., Aikin, K., Aoki, S., Apadula, F., Baier, B., Bergamaschi, P., Beyersdorf, A., Biraud, S. C., Bollenbacher, A., … and Zimnoch., M.: *CarbonTracker CT2019B*, NOAA Global Monitoring Laboratory, https://doi.org/10.25925/20201008, 2020.
- Liu, J. and Bowman, K.: Carbon Monitoring System Carbon Flux Land Prior L4 V3, Greenbelt, MD, USA, Goddard Earth Sciences Data and Information Services Center (GES DISC), Last access: 17 June 2024, 10.5067/1XO0PZEZOR1H, 2024.
- Ott, L.: GEOS-Carb CASA-GFED 3-hourly Ecosystem Exchange Fluxes 0.5 degree×0.625 degree V3, Goddard Earth Sciences Data and Information Services Center (GES DISC), Greenbelt, MD, USA, https://doi.org/10.5067/VQPRALE26L20, 2020.

Please also review the English of the sentence "These findings agree with Gaubert et al. (2023), which showing most of the inverse models in v10 OCO-2 MIP have significant errors because of potential positive biases in OCO-2 XCO2 measurements for this region.".

[Response] We revised the sentence (L364-365 in the revised manuscript) as follows:

"These findings agree with Gaubert et al. (2023), which shows that most inverse models assimilating OCO-2 XCO$_2$ retrievals tend to overestimate the net carbon sources in this region because of potential positive biases in the OCO-2 retrievals.